# Biomimetic Polarized Light Navigation Sensor: A Review

**DOI:** 10.3390/s23135848

**Published:** 2023-06-23

**Authors:** Shunzi Li, Fang Kong, Han Xu, Xiaohan Guo, Haozhe Li, Yaohuang Ruan, Shouhu Cao, Yinjing Guo

**Affiliations:** 1College of Electronic and Information Engineering, Shandong University of Science and Technology, Qingdao 266590, China; 202183130018@sdust.edu.cn (S.L.); 202183130031@sdust.edu.cn (H.X.); 202183130062@sdust.edu.cn (H.L.); 202183130001@sdust.edu.cn (Y.R.); 202183130024@sdust.edu.cn (S.C.); 2College of Electrical Engineering and Automation, Shandong University of Science and Technology, Qingdao 266590, China; 202181080004@sdust.edu.cn; 3School of Information Science and Engineering, Shandong University, Qingdao 266237, China; 201920332@mail.sdu.edu.cn

**Keywords:** polarized light navigation, point source polarized light sensor, imaging polarized light sensor, micro nano processing technology polarized light sensor

## Abstract

A polarized light sensor is applied to the front-end detection of a biomimetic polarized light navigation system, which is an important part of analyzing the atmospheric polarization mode and realizing biomimetic polarized light navigation, having received extensive attention in recent years. In this paper, biomimetic polarized light navigation in nature, the mechanism of polarized light navigation, point source sensor, imaging sensor, and a sensor based on micro nano machining technology are compared and analyzed, which provides a basis for the optimal selection of different polarized light sensors. The comparison results show that the point source sensor can be divided into basic point source sensor with simple structure and a point source sensor applied to integrated navigation. The imaging sensor can be divided into a simple time-sharing imaging sensor, a real-time amplitude splitting sensor that can detect images of multi-directional polarization angles, a real-time aperture splitting sensor that uses a light field camera, and a real-time focal plane light splitting sensor with high integration. In recent years, with the development of micro and nano machining technology, polarized light sensors are developing towards miniaturization and integration. In view of this, this paper also summarizes the latest progress of polarized light sensors based on micro and nano machining technology. Finally, this paper summarizes the possible future prospects and current challenges of polarized light sensor design, providing a reference for the feasibility selection of different polarized light sensors.

## 1. Introduction

The essence of light is a transverse electromagnetic wave in a specific spectral range radiated from the light source. The vibration directions of the electric field vector E and the magnetic field vector H are both perpendicular to the transmission direction [1]. Polarization (POL) is the unique property of light as a transverse electromagnetic wave, reflecting that the vibration direction of the optical wave electric vector does not have symmetry with the propagation direction of light. Polarization information is valued by people in the fields of navigation [2,3,4,5], medicine [6,7], and remote sensing [8,9,10]. Navigation technology plays an important role in people’s daily lives, and it is indispensable for both daily travel and industrial transportation. Current mainstream navigation modes include Global Navigation Satellite System (GNSS), Inertial Navigation System (INS), astronomical navigation, ground-based radio navigation, and geophysical navigation. According to different applicable scenarios, these navigation methods play an important role in various application fields, but it is undeniable that each navigation method has certain limitations.

Satellite navigation is currently the most widely used non-autonomous navigation method with high-precision positioning information. It can provide users with all-weather, high-precision positioning; navigation; and timing services. However, the GNSS is susceptible to natural or human interference, especially in weak satellite signals such as jungle and underwater environments, which cannot provide effective navigation information [11]. The INS is a commonly used autonomous navigation method for underwater AUVs, which has the characteristics of high short-term accuracy, good stability, good concealment, and less susceptibility to external interference. However, due to the generation of navigation information through path integration, its errors accumulate over time and require timely error correction [12]. The astronomical navigation system is an autonomous navigation system that uses the position information related to celestial bodies and time to calculate the heading, attitude, and position of aircraft. It is often used in the field of aerospace and aviation. Its navigation error does not accumulate over time and has good concealment. Its navigation accuracy depends on the accuracy of optical sensors, and its navigation performance is ideal when applied in high altitude and space with thin air, but its obvious disadvantage when applied near the Earth is that it is severely affected by cloud cover and climate conditions [13]. The land-based radio navigation system uses the Doppler frequency shift effect to measure the carrier speed to achieve the purpose of positioning and navigation. It has the characteristics of small size, simplicity, and low cost. However, because it relies on electromagnetic wave propagation in space, it has the disadvantages of poor anti-interference ability and limited coverage [14]. Geophysical navigation technology is a technology that utilizes the inherent attributes of the Earth for navigation, including geomagnetic navigation, terrain matching navigation, and gravity gradient navigation. Geophysical navigation technology has advantages such as good concealment, no time and regional limitations, strong anti-interference ability, and no error accumulation. For geomagnetic navigation, accurate geomagnetic field models and complete geomagnetic databases are the foundation for achieving high-precision geomagnetic navigation. Therefore, the improvement of magnetic measurement equipment and effective algorithm compensation for the impact of interference magnetic fields are still the future development direction [15]. Terrain matching navigation is an autonomous auxiliary navigation method suitable for low altitude aircraft, sea exploration, cruise missiles, and other fields. It has high requirements for equipment and terrain data, and is currently only applied in the military field, and cannot be used at sea, on plains, and high altitudes above 300 m [16]. Gravity gradient navigation is currently widely used in underwater vehicles, often as an auxiliary means of INS. However, the mass and volume of gravity gradient instruments are relatively large, which cannot meet the development needs of miniaturization of underwater vehicles [17]. From these navigation methods, most of the high-precision navigation methods rely on radio survival, and once signal rejection or human interference occurs, it will have a very adverse impact on the navigation results. Although navigation methods based on natural characteristics are not easily damaged by human factors, the current accuracy achieved cannot fully meet the navigation needs of military and civilian applications. Therefore, studying a navigation method based on natural characteristics with high accuracy is of long-term significance.

Polarized light navigation is a new autonomous navigation method developed by imitating the eye perception structure of insects such as sand ants [18,19], locusts [20], cuttlefish [21], bees [22], and North American monarch butterflies [23] that navigate by detecting polarized light in the sky. Bionic polarized light navigation is an autonomous navigation technology that uses a stable sky polarization distribution mode as the signal source, and it has the advantages of strong autonomy, strong anti-interference ability, and no error accumulation over time [24]. At present, there are many kinds of sensors that simulate and imitate the insect compound eye structure to detect and identify sky polarized light and process signals, mainly including point source sensors and imaging sensors. Point source sensors can be divided into basic point source sensors and point source sensors used in integrated navigation. Both of them have their own advantages and disadvantages. For example, although the basic point source sensor is simple in structure and small in size, its stability is poor. Although the point source sensor used in integrated navigation can improve the navigation accuracy, it has the problem of large size and high cost. The imaging sensor can be divided into time-sharing imaging sensor, real-time amplitude splitting sensor, real-time aperture splitting sensor, and real-time focal plane splitting sensor. The classification, system composition, references, advantages, and disadvantages of various sensors are shown in Table 1. From the published research results, it can be seen that the use of sky polarized light to achieve navigation has a broad application prospect. This paper summarizes the research progress of polarized light sensors. Compared to literature [25], which categorizes and reviews polarization navigation sensors based on research teams and years, this article focuses more on the technical and structural aspects of the sensors. Reference [26] introduces the research progress of biomimetic polarization navigation technology from two aspects: the distribution of atmospheric polarization pattern maps and polarization navigation sensors. This includes three angles: polarization field distribution theory, polarization distribution testing, and polarization navigation technology. The article focuses on the distribution of atmospheric polarization pattern maps and only provides a brief introduction to the structure of polarization navigation sensors. Based on the research results of bionics, the literature [27] expounds the polarization vision detection mechanism of many kinds of organisms, and then, aiming at the application direction of autonomous navigation, summarizes the research progress of polarized light navigation technology from the distribution of sky polarization mode and the design and application of polarized light navigation sensors, without summarizing and classifying polarized light sensors. Reference [28] provides an overview of photodiode sensors with linear thin film polarizers, camera-based polarization navigation sensors, focal plane splitting sensors, and a comparison of the three types of sensors in terms of polarization navigation sensors. However, due to the limited space and age, there are certain limitations in comprehensiveness. Reference [29] reviews the navigation strategies for beetle linear orientation and the neural network foundation for navigation orientation, without providing a detailed description of navigation sensors in practical applications. In 2012, Salmah B. Karman et al. reviewed the principles, structures, and algorithms of polarization navigation sensors with linear thin film polarizers, camera-based polarization sensors, and focal plane splitting polarization sensors in reference [30]. They did not provide a comprehensive review of sensors used for integrated navigation and micro/nano processing technology sensors developed in the past decade. This paper compares and analyzes the polarized light navigation mechanism, point source sensor, imaging sensor and sensor based on micro/nano processing technology, and their refinement branches, providing a reference for the feasibility selection of different polarized light sensors.

## 2. Biomimetic Polarized Light Navigation in Nature

Biologists have continuously discovered that some insects [97], birds [98], fish [99], and others in nature can navigate by sensing polarized light in the sky to assist in their foraging, homing, and migration behaviors. After 3.5 billion years of evolution, organisms have gradually formed tissues, structures, and organs that adapt to their growth environment. For example, sand ants can reach several hundred meters away from their nests when foraging but can return to their nests in almost a straight line (thick line in the picture) [3], as shown in Figure 1.

Based on the summary of biologists, it can be concluded that the sand ant uses the autonomous navigation method of optical flow when departing, which obtains global vector information from the polarized light perception structure of the eye, records the directional distance of each step with an auto-odometer, and then adds vector superposition of the directional information and distance information to obtain the current position relative to the initial position. Anatomical studies have shown that most organisms with polarized light vision or sensitivity to polarized light rely on some unique structures in their eyes, namely, the small eyes arranged in an orderly manner in the dorsal rim area (DRA), as shown in Figure 2.

Each small eye of the sand ant DRA contains eight mutually orthogonal neural rods. The structural diagram of the sand ant neural rods is shown in the Figure 3a, and the arrangement diagram of the neural rods is shown in the Figure 3b. Each pair of orthogonal neural rods constitutes polarization sensitive neurons (POL neurons). Sand ants rely on the retina layer in the compound eye to sense polarized light, and the central nervous system processes the perception information. Orthogonal unidirectional photoreceptors form neural rods that can sense polarized light in the same direction as them.

When the sand ant senses polarization information, each channel of the neural stem generates a signal that is transmitted to the neuron for processing. After processing, the polarization information outputs a signal that is close to a sine curve. It is precisely inspired by the wonderful polarized visual navigation ability and special visual organs of these organisms that domestic and foreign scholars have begun to explore and experiment in the design of biomimetic polarized light sensors.

Polarized light navigation is a new autonomous navigation method developed by imitating the eye perception structure of insects such as sand ants, locusts, cuttlefish, bees, and North American monarch butterflies that navigate by detecting polarized light in the sky. African dung beetles [100] and beetles [101,102] can even use weakly polarized moonlight for navigation. The polarized visual navigation of some animals is shown in Figure 4. After repeated calculations and verifications by scientists, it has been found that at specific times and locations throughout the day, there exists a stable atmospheric polarization distribution pattern in the entire zenith region, as shown in Figure 5. Due to the effect of particles of different sizes on light, under ideal conditions of clear and cloudless skies, the scattered particles in the atmosphere are much smaller than the wavelength of light. Therefore, we usually use the Rayleigh scattering model to characterize the scattering characteristics of sunlight. In the first-order scattering Rayleigh model, the distribution of polarized light in the sky is regular and stable. With the spherical center O of the ground as the location of the observer, the intersection point between the observer and the celestial sphere is zenith Z, the sun meridian axis passes through the sun S and the zenith, and the polarization mode distribution of the whole sky is mirror symmetry with the sun and the anti-sun meridian (SM-ASM) as the axis. The dotted circles around the sun in the figure represent the distribution characteristics of polarized light in the sky, the thickness of the dotted lines represents the degree of polarization of the corresponding points, and the direction of the dotted lines represents the vector vibration direction of polarized light at the corresponding observation points. Due to the stable characteristics of the polarization distribution under the Rayleigh scattering model similar to the Earth’s geomagnetic field, it is possible to use atmospheric polarized light for navigation [103].

## 3. Biomimetic Polarized Light Navigation Mechanism

Before entering the atmosphere, sunlight is non-polarized natural light. After entering the atmosphere, because atmospheric molecules, aerosols, and other particles in the atmosphere scatter sunlight, they will change the polarization state of light, resulting in sky polarized light [104]. Sky polarized light forms a stable polarized light distribution pattern in the macro, which contains rich directional information, such as meridian, degree of polarization (DOP), and angle of polarization (AOP). The degree and angle of polarization represent the degree of polarization of light and the direction of vibration of light (E vector direction), respectively. Sand ants, bees, and other insects detect the polarization information of the sky to achieve autonomous navigation [19]. When a beam of light passes through a polarizer, the relationship between the outgoing light and the incoming light can be described by the state transfer matrix Muller matrix [14]. Polarized light sensor can be divided into point source polarized light sensor based on the POL neuron model and imaging polarized light sensor according to measurement means. The two measuring principles are basically the same, and both are described by the Stokes vector method. The Stokes vector has four components, namely, $S={\left[I,Q,U,V\right]}^{T}$, which are defined as follows:(1)$$\begin{array}{l} I=E_{x}^{2}(t)+E_{y}^{2}(t)\\ Q=E_{x}^{2}(t)-E_{y}^{2}(t)\\ U=2E_{x}^{2}(t)E_{y}^{2}(t)\cos[\delta_{y}(t)-\delta_{x}(t)]\\ V=2E_{x}^{2}(t)E_{y}^{2}(t)\sin[\delta_{y}(t)-\delta_{x}(t)] \end{array}$$
where I is the total light intensity; Q and U are, respectively, two mutually perpendicular linear polarization components; V is the right-handed circularly polarized light; and $E_{x}$ and $E_{y}$ are, respectively, the components of the electric vector intensity of light in the two vertical axes. Since the sky polarized light is mainly linear polarized light and there is little circular polarized light, it is considered that V=0.

There are multiple different channels (at least three) in the polarized light sensor. Each channel is composed of linear polarizers and photoreceptors. By detecting the light intensity response of each channel, the measurement of degree of polarization and angle of polarization can be realized. Combined with the description of Stokes vector method, the degree and angle of polarization can be defined as
(2)$$\phi =\frac{1}{2}\arctan(\frac{U}{Q})$$
(3)$$d=\frac{\sqrt{Q^{2}+U^{2}}}{I}$$

Define that the included angle between the measured main axis and the selected axis x of the Stokes vector is $\alpha_{j}$, which is called the installation angle of the polarizer. It can be obtained through prior calibration. Suppose that the Stokes vector of an incident beam is S, the state transfer matrix of the polarizer is M, and the Stokes vector of the outgoing beam after passing the polarizer is $S^{\prime }$, then
(4)$$S^{\prime }=MS$$

Namely,
(5)$$\left[\begin{array}{c} I^{\prime }\\ Q^{\prime }\\ U^{\prime }\\ V^{\prime } \end{array}\right]=\left[\begin{array}{cccc} m_{00} & m_{01} & m_{02} & m_{03}\\ m_{10} & m_{11} & m_{12} & m_{13}\\ m_{20} & m_{21} & m_{22} & m_{23}\\ m_{30} & m_{31} & m_{32} & m_{33} \end{array}\right]\left[\begin{array}{c} I\\ Q\\ U\\ V \end{array}\right]$$

Mueller matrix is
(6)$$M=\frac{1}{2}\left[\begin{array}{cccc} 1 & \cos2\alpha_{j} & \sin2\alpha_{j} & 0\\ \cos2\alpha_{j} & \cos^{2}2\alpha_{j} & \cos2\alpha_{j}\sin2\alpha_{j} & 0\\ \sin2\alpha_{j} & \cos2\alpha_{j}\sin2\alpha_{j} & \sin^{2}2\alpha_{j} & 0\\ 0 & 0 & 0 & 0 \end{array}\right]$$

The light intensity accepted by each channel is
(7)$$I(\alpha_{j})=E_{x}^{2}\cos^{2}(\alpha_{j})+E_{y}^{2}\sin^{2}(\alpha_{j})+E_{x}E_{y}\sin(2\alpha_{j})$$

Simplification:(8)$$I(\alpha_{j})=\frac{1}{2}[I+Q\cos(2\alpha_{j})+U\sin(2\alpha_{j})]$$

It can be seen from (8) that at least three different measurement equations are required, that is, Stokes vector parameters can be calculated, and then the measurement in-formation of polarized light can be obtained through (2) and (3).

## 4. Research Progress of Bionic Polarized Light Navigation Sensor

### 4.1. Point Source Polarized Light Sensor Based on POL Neuron Model

#### 4.1.1. Basic Point Source Polarized Light Sensor

In 1997, Lambrinos et al. built a six-channel polarized light sensor simulating the structure of cricket’s eyelet based on the POL neuron principle and installed this sensor on the Sahabot robot, initially realizing the navigation function [2]. On this basis, in 2000, Lambrinos et al. proposed a synchronization model to analyze and combine the signals of two POL cells to eliminate the degree of polarization and used the POL compass in the path integration system [3]. In 2012, Chahl et al. designed a bionic polarization detection device according to the polarization vision mechanism of dragonfly and applied it to small unmanned aerial vehicles. The calibrated device is comparable to the solid compass in accuracy, but there are problems such as discontinuous heading angle output and low integration [31]. In 2019, Gkanias et al. established a model to directly estimate the solar azimuthal angle by dealing with the polarizability of different parts of the sky in different directions. In addition, they proposed a method to correct for sensor array tilt, which can be used as an input for insect path integrated neural [32].

Chu’s research group made the following research: In 2008, Chu et al. built a three-channel sensor based on the polarization mechanism of the compound eye of the sand ant and gave the photoelectric structure of the silicon photodiode and the circuit based on the ARM microprocessor [4]. The sensor results showed that the nonlinear error was less than 0.0233%, and the worst angle output error was within ±0.2°. In 2009, Zhao et al. designed a six-channel polarized light sensor and proposed a new angle output algorithm, which can completely avoid the overflow problem of division by zero [33]. In 2019, Wang et al. proposed a biological polarization sensor with a plano convex lens [34]. The role of the plano convex lens is to narrow the field of vision and block stray light. In addition, they proposed two calibration algorithms, namely, “central symmetry algorithm” and “discontinuous algorithm”, to improve the accuracy of the sensor. Finally, the sensor shows indoor accuracy of ±0.009° and outdoor accuracy of ±0.018° under clear sky conditions, as shown in Figure 6.

In 2019, Dupeyroux et al. proposed a two-pixel polarization sensitive visual sensor. The spectral sensitivity of this sensor is within the ultraviolet range. It is a minimalist sensor, as shown in Figure 7a [35]. The sensor is embedded in the hexapod robot and five algorithms for calculating AOP are compared: Stokes–Mueller algorithm [36], Sahabot algorithm [3], matrix algorithm adapted from the Sahabot method, extended algorithm (including the N-dimension extension of Sahabot), and neuron-like AntBot algorithm based on the neuron model [37]. After comparing the polarization angle errors of the five algorithms, it was concluded that the last two algorithms are better than the first three algorithms, and the average angular error is only 0.62° ± 0.40° and 0.69° ± 0.52°. The advantage of this sensor is that only two photodiodes can be used to obtain the azimuth information, which can reduce the complexity of calculation. The disadvantage is that the spectral sensitivity is only within the ultraviolet range, and the fact that other insects are sensitive to blue and green light is not taken into account. Later, Dupeyroux et al. designed a single pixel waterproof ultraviolet (UV) polarized light compass [18] and encapsulated the POL device in a customized epoxy resin shell. The experimental results show that under rainy conditions and in freshwater at lower depths, there is almost no effect on AOP, and the homing error is less than 0.7% of the entire trajectory. The general overview of the point source polarized light sensor is shown in Table 2.

#### 4.1.2. Point Source Sensor for Integrated Navigation

In the field of integrated navigation, Dupeyroux et al. proposed a hexapod robot equipped with two UV polarized light sensors [38], as shown in Figure 7b, and then combined it with GPS navigation to simulate the outdoor homing behavior of sand ants. The experimental results show that the integrated navigation system using path integration strategy can improve its robustness and efficiency, and the average homing error is as low as 0.67%. Guo’s research group produced the following research: In 2020, Du et al. designed a six-channel vertical polarized light sensor and proposed a static autonomous initial alignment algorithm [39]. They combined the polarization measurement error equation with the error equation of the strapdown inertial navigation system (SINS) to achieve autonomous and fast initial alignment. The simulation results show that the alignment accuracy of the integrated navigation error model was improved by more than 90%, and the time was shortened by more than 60%. In 2021, Liu et al. proposed a bionic attitude and heading reference system [40], which is composed of a compound eye polarization compass and an inertial measurement unit. The polarization compass is a noncoplanar polarization type structure with multi-directional observation channels. As shown in Figure 7c, it can adaptively select the polarization angle and obtain the polarization vector. They proposed a calibration model to compensate for installation errors. Finally, the root mean square error of heading angle is 0.14° under clear sky conditions and 0.42° under cloudy conditions.

Aiming at the problem that electromagnetic interference affects the navigation and positioning accuracy, in 2020, Yang et al. proposed a bionic global autonomous positioning system based on polarization skylight [41], and first proposed the algorithm to determine the overall maximum DoP (MDoP), which can avoid the impact of E vector error on solar elevation measurement. They considered the sun elevation instead of the traditional sun azimuth and used the sun elevation difference algorithm to derive the position. In the case of magnetic interference, the positioning performance was improved, and the positioning error was within a few kilometers, with the latitude error within 0.113° and longitude error within 0.082°. In the same year, Jian Yang et al. proposed a bionic polarization-based attitude and heading reference system assisted by inertial sensors [42], which solved the problem of determining and resolving the three-dimensional polarization heading. At the same time, the system measurement model also coupled the attitude and heading cumulative errors of the inertial navigation system. The experimental results show that this system can solve the ambiguity problem of the polarization heading in the case of electromagnetic interference, and the accuracy of polarization angle was less than 0.2°. In 2021, Zhang et al. designed a hemispherical polarization navigation sensor to detect polarized sky light information (degree of polarization and E-vector) [43]. Each polarization sensor unit is distributed on the hemispherical surface in the form of an array as an independent observation point. The coordinate transformation relationship of each polarization sensor unit is determined by the sensor structure. Even if part of the sky is blocked, polarized sky light information in different directions can be observed.

Chu’s research group produced the following research: In 2009, Chu and others first used polarized light sensors to obtain the absolute orientation information, then mapped the information to the robot through a fuzzy logic controller and applied the polarization sensor and wheel encoder to the robot’s automatic navigation [44]. In 2018, Zhi et al. applied the polarization sensor to the conventional attitude determination system. Using the vertical relationship between the E vector direction of the centerline polarized light and the sun vector direction of the atmospheric polarization distribution mode, they derived the heading angle function and proposed an extended Kalman filter with the quaternion differential equation as the dynamic model, which was applied to the flight control of an Unmanned Aerial Vehicle (UAV) [45]. The results show that the accuracy of the polarization sensor can be limited to 0.2°. As shown in Figure 7d, this system has the advantage of being able to work in a magnetic interference environment, but further optimization is needed in the calculation of Jacobian matrix and measurement noise model. In 2015, Wang et al. proposed a combined timing positioning device based on polarized light and geomagnetic field, which consists of two polarized light sensors, a three-axis compass, and a computer. The advantage of this equipment is that it works independently of the manual signal source, has no error accumulation, and can directly obtain the position and orientation. When the pitch and roll angles are within 85°, the accuracy of the polarized light sensor is within ±0.2°. The problem is that only two E vectors can be measured [46], as shown in Figure 8a. To solve this problem, Wang et al. developed a new sensor device for measuring five E vectors, as shown in Figure 8b [47], and proposed an orthogonal vector algorithm using the redundancy of the single scattering Rayleigh model. Five sensors are fixed on five planes, which can simultaneously measure more E vectors and significantly increase the robustness of the equipment. The average deviation of the measured solar vector angle is about 0.242°. The disadvantage of this equipment is that if the included angle between E vectors is close to 0°, the result will be very inaccurate, and the performance of the equipment when it is not horizontally aligned should be further improved. In 2022, Li et al. applied polarized light navigation to small UAVs and designed an integrated navigation system composed of polarized light sensors, Micro Inertial Measurement Unit (MIMU), and Global Positioning System (GPS) [48]. They proposed a polarization angle output algorithm and designed a tilt correction algorithm to solve the problem that the polarization angle error increases when the Unmanned Aerial Vehicle (UAV) tilts. The system can obtain the complete navigation parameters of the small UAV and meet the accuracy and reliability requirements of the small UAV; the indoor polarization angle error of the polarized light sensor is about 0.2°, and the outdoor polarization angle error is less than 2°.

Hu’s research group produced the following research: In 2021, Zhou et al. proposed a bionic multi-sensor navigation and control system, which is composed of a pixel polarization sensor, a micro inertial measurement unit (MIMU) and a monocular camera, as shown in Figure 8c. They proposed a multi-sensor joint calibration algorithm and an adaptive integration algorithm, which can solve the problems of ambiguity of polarization direction and cumulative error. The experimental results show that the root mean square error of the calibrated azimuth was 0.014° [49]. In 2021, Xie et al. proposed a bionic navigation system consisting of a three-axis magnetometer, a monocular camera, a micro inertial measurement unit, and a polarization camera. The dead reckoning algorithm is used to integrate the monocular camera and micro inertial measurement unit into the visual inertial odometer, and then the absolute orientation of the magnetometer is combined with the relative orientation of the visual inertial odometer to achieve navigation. The proposed navigation system is applied to the vehicle, the root mean square error of the sensor’s position error is 0.78% of the total driving distance in dataset A, and the root mean square error of the azimuth error is 0.9°. In dataset B, they are 1.31% and 0.9°, respectively [50]. In 2022, Chen Fan et al. proposed a bionic multi-sensor navigation system with a polarized light compass and visual position recognition [51], and proposed an optimal orientation algorithm and two-dimensional visual position recognition technology. This system can establish bionic and robust correlation between motion information and sensor input and use reliable position and heading constraints to improve navigation performance. The results show that the position error is convergent. The root mean square error of the navigation system’s position on the straight flight dataset is 0.21%, and the root mean square error on the square flight dataset is 0.07%. In 2022, Ying Fan et al. proposed robust bionic polarized sky light orientation algorithms [52] and polarization mode consistency algorithms (MCOPPs) for cloudy weather, and proposed a two-step optimization framework for effectively selecting interior points. The experimental results show that the orientation accuracy of MCOPP in outdoor static experiments is within 0.5°, and that in dynamic automobile experiments is within 0.7°.

For attitude determination, in 2020, Liang et al. analyzed in detail the rotational symmetry and plane symmetry of the Rayleigh sky model [53], and theoretically proved that the Rayleigh sky model only contains a single sun vector information, which only contains two independent scalar pose information, so it is impossible to obtain three Euler angles in real time at the same time. In 2021, Hu et al. applied the advantages of the polarization navigation sensor, such as autonomy and no cumulative error, to the error correction in the inertial navigation system. As shown in Figure 8d, in order to eliminate the negative impact of polarization navigation, they proposed an attitude angle adaptive partial feedback algorithm to adjust the error correction degree of inertial navigation in an adaptive form. This method can improve the accuracy of three-dimensional attitude estimation [54]. In 2022, Zhao et al. analyzed the heading error caused by pitch angle and roll angle during flight of UAV guided by polarized light [55]. A new directional error modeling and compensation algorithm for polarization compass attitude change is established by using the Gated Recursive Unit (GRU) neural network, as shown in Figure 8e. Experimental results show that the proposed algorithm performs well. The root mean square error of the heading error model is 0.5218°. In 2022, Zhao et al. proposed an INS/GNSS/polarized light integrated navigation system based on volume Kalman filtering, which has higher navigation accuracy compared with traditional integrated navigation systems [56]. See Table 3 for an overview of point source sensors applied to integrated navigation.

### 4.2. Imaging Polarized Light Sensor

The imaging sensor can be divided into a time-sharing imaging sensor and real-time imaging sensor according to the imaging system of the sensor [24], and the real-time imaging sensor can be divided into an amplitude-splitting sensor, aperture splitting sensor, and focal plane splitting sensor. Compared with traditional navigation and positioning devices and systems, imaging sensors are lighter in shape, lighter in weight, higher in integration, better in photosensitivity, and stronger in autonomy [105]. Compared with the point-source sensor, the polarized light information collected from the sky is more abundant. Because of the imaging sensor, the information contained in the image can also make us more objective and detailed understanding and analysis of the sky polarized light. Moreover, the imaging sensor can also work in various modes, with better robustness and adaptability, and can better output the navigation angle. On the premise of meeting the requirements of real-time navigation and positioning, it can also perform complex post-processing data [105]. Common image sensors are CMOS and CCD. The CCD sensor is a very integrated sensor; this sensor photo reading speed is fast, and CMOS is known for its small size and light weight. CMOS sensor power consumption is small and has long service life, but also has a very high sensitivity, so CMOS is the mainstream device of image sensors. Time-sharing sensor imaging sensor refers to obtaining different polarization information in discrete time by continuously changing the state of the polarization device [106]. A time-sharing imaging sensor cannot capture polarization pictures of multiple polarization angles at the same time. Its main feature is that images of multiple polarization angles are captured in different time periods. Each acquisition of polarization information of an object requires multiple rotations of the polarizer or wave plate in front of the lens. Similarly, the real-time imaging sensor can image the object in real time, without rotating the polarizer, and can obtain multiple polarization images with different polarization states by shooting the target once [107]. A real-time imaging system can be divided into amplitude type, aperture type, and focal plane type according to the different structural light path of the system.

#### 4.2.1. Sensor Based on Time-Sharing Imaging System

The key of the time-sharing imaging system in acquiring a polarization image is that the polarizer images the target from different angles to obtain the polarization image, which is suitable for stationary targets. In 1977, Walraven et al. proposed a polarization imaging system. The polarizer in the linear polarization imaging sensor rotates in front of the film camera, digitizes the developed film, and calculates the linear Stokes vector element [57]. In 1999, P. Bishop et al. developed a multispectral imaging polarimeter, as shown in Figure 9a. This instrument achieved multiple images of the scene at multiple different rotation positions of the wave plate through rotating wave plate and linear polarizer [58]. In 2006, J. Push et al. used a Liquid Crystal Variable Retarder (LCVR) to establish and calibrate the imaging Stokes vector polarimeter. Using two temperature-controlled LCVRs and a fixed linear polarizer, four images can be shot at different times, and the measurement time of the four images varies between 0.3 s and 1.2 s [59]. However, the liquid crystal variable retarder has problems such as limited spectral range and slow frame rate. In 2009, Miyazaki et al. proposed to use the angled 180° field of view fish-eye lens to capture the sky polarization image, which can be taken by rotating the polarizer, as shown in Figure 9b. However, manually rotating the polarizer and tilting the camera will cause large errors [60]. In 2014, Wang et al. proposed a sensor composed of a CCD camera, fish-eye lens, and linear polarizer to measure the polarization of sky light, deduced a polarized light parameter estimation algorithm based on three images, and introduced the least square estimation into the measurement of sky light polarization image. Finally, the polarization mode of the sky light obtained is compared with the single scattering Rayleigh model, and the conclusion is that 90% of the deviation is less than 5°, and 40% of the deviation is less than 1°. Figure 9c shows the sensor system shown [61]. In 2020, Cai et al. proposed a new novel hue-based color mixing (HBCM) model, which can more accurately calculate polarization information and direction information [62]. Although they implemented the HBCM method based on color mixing technology to avoid environmental interference and improve the AOP mode, they did not obtain more polarization information and then calculate the heading angle and navigation information in the urban environment with more complex environment. Figure 9d shows the sensor system.

The sensor based on the time-sharing imaging system mostly relies on motor or manual to drive the polarizer to rotate. This kind of polarization imaging system has the advantages of simple structure, low price, and convenient construction. However, when shooting images, it is necessary to keep the position of the camera and the object fixed, and the shooting time is too long to be used for shooting dynamic objects and external complex environments. The change of the light intensity of the external light source will affect the accuracy of the polarized image, and the rotation angle and displacement of the polarizer will also cause projection errors [24]. To solve these problems, researchers have developed a real-time polarization imaging system.

In addition to daytime polarized light navigation, there is also research on moonlight night sky polarization, which is due to noise effects and model uncertainty. Although it has been found that insects such as scarabs can perform polarized navigation under moonlight at night [63], the practical application of nighttime polarization orientation has only occasionally appeared in previous studies on polarized light. In 2001, Gal et al. used a 180° imaging polarizer to measure the polarization degree and angle of the clear night sky under a full moon every half an hour, finding that if the lunar zenith angle is the same as the solar zenith angle, the conclusion is that the neutral point position is the same [64]. In 2003, Dacke et al. found that beetles use moonlight polarization to locate their position by placing polarization filters on them [65]. In 2011, Kyba et al. conducted a study on the polarization signal of the moon caused by urban lighting pollution. The results showed that due to the inconsistent AOP between skylight and scattered moonlight, urban skylight reduced the natural polarization signal of the moon, and as the position of the moon rose, the polarization degree decreased [66]. In 2016, Jun Tang et al. proposed a new method for calculating polarized sky light imaging compass information based on the Pulse Coupled Neural Network (PCNN) algorithm. The experimental results showed that the accuracy of the compass was 0.1805° on clear weather, and 0.878° on weak lunar polarization information sources [67]. In 2022, Yang et al. used a cryogenic camera and a fisheye lens to form a time-sharing imaging system to measure the polarization of the moon sky at night. The composition of the imaging system is shown in the figure. Based on the statistical characteristics of the polarization angle error, they proposed a probability density estimation method that can determine the heading and used this method to conduct robust modeling of the error distribution of AOP. The performance of the sensor results showed that in the clear night sky, the standard deviation for heading estimation is 0.32°. In a cloudy night sky, the standard deviation is 0.47° [68]. In addition, references [63,69,70] also studied the effects of the moon on the polarization patterns of celestial bodies, such as DOP and AOP, indicating that a smaller lunar illumination fraction corresponds to a lower DOP. Although the above research provides effective evidence for us to use nighttime polarized light during navigation, compared to daytime polarization modes, the polarization modes generated by the moon are very complex, and the low light environment at night poses challenges to the detection of polarization modes. Polarization modes are often disrupted by noise during acquisition or transmission. The application of nighttime polarization signals as an effective navigation technology still faces many challenges.

#### 4.2.2. Divided Amplitude Sensor Based on Real-Time Imaging System

In order to detect moving objects, scientific researchers have invented the amplitude-splitting polarization imaging. The light enters the objective lens, and the output light of the objective lens passes through a series of polarizing beam splitters and retarders and becomes three or more polarized light waves in different directions [108]. The polarized light waves in different directions converge to form images on the focal plane of different image sensors. The last three or more polarized pictures in different directions are combined to obtain a complete polarized image [105]. Most of the biomimetic polarization sensor schemes use linear polarizers as polarizers, which will inevitably lead to orthogonal errors between polarizers that are orthogonal to each other. In order to solve this problem, in 2010, Azzam et al. used a spectroscope to divide the incident light into three beams. The beams passed through a 1/4 wave plate and a 1/2 wave plate and then were imaged on three detectors to obtain different polarization information [71]. The device can simultaneously detect pictures with multiple polarization angles. The optical path of the beam splitting is on the same axis. Therefore, there is no deviation error between the obtained polarization pictures, but the measurement system has obvious shortcomings in structure. The incident beam is divided twice, and the light brightness detected by each detector is low, and the brightness of imaging on different detectors is inconsistent. Because the system uses multiple cameras and contains multiple light splitters and polarization devices, the volume of the fractional amplitude imaging system is usually large. In 2016, Ben et al. proposed a system model using polarization beam splitter instead of polarizer [72]; as shown in Figure 10a, the root mean square error of this model is about 0.026°, and the average absolute error is 0.02°. In addition, they put forward a calibration algorithm. After testing, they concluded that the calibration algorithm reached the equivalent accuracy of the nonlinear least squares method. Although Ben et al. proposed a system model based on the polarization beam splitter, this model is still a simplified theoretical model. In practice, the sensor will be affected by the incident angle, wavelength sensitivity, parameter uncertainty, etc. The way in which to compensate the error of the model needs further research.

In 2018, Yang et al. designed a polarized light sensor based on the polarization beam splitter. Aiming at the problem of optical path coupling and extinction ratio inconsistency between the transmission beam and reflected beam of the polarization beam splitter, a new sensor model was established, and a coupling coefficient was introduced to solve this problem [73]. A new algorithm was proposed, and the calibration test was carried out using the unscented Kalman filter calibration method. In the sensor design, in order to ensure that the incident light is perpendicular to the polarization beam splitter, they designed a traction tube, as shown in Figure 10b. This kind of traction tube is essentially a lens cover, which can avoid the influence of external stray light on the polarization beam splitter, and then can decompose the incident non-polarized light into two orthogonal polarized lights. Through error analysis, calibration, and compensation, the accuracy can reach 0.18°. In 2019, Yang et al. proposed a new sensor error model [74], and the test equipment is shown in Figure 10c. The model introduces the sensor coefficient error; establishes the polarization angle error model and the polarization degree error model; and then classifies the multi-source interference, including the sky conditions, the spectral component of the incident sunlight on the sensor, and the inconsistency of the extinction ratio of the spectrometer. The sensor coefficient is calibrated using the least squares optimization algorithm. The root mean square error accuracy of the polarization angle of the sensor is approximately 0.12°.

Although the position and other errors of the amplitude-divided sensor are small, the optical structure is complex and the fabrication is difficult, and the light intensity has been greatly weakened before reaching the imaging unit. Moreover, the split-amplitude polarization detection sensor has a multi-path structure. When obtaining the polarization information of the target, the system is required to use multiple images obtained at the same time to calculate the results. Under this requirement, the multiple images should achieve a good registration relationship, which can output accurate polarization information. Therefore, when using the split-amplitude sensor, an image registration method that can well adapt to the polarization detection sensor is needed [109].

#### 4.2.3. Aperture Dividing Sensor Based on Real-Time Imaging System

The split-aperture real-time imaging system is similar to the split-amplitude imaging system in an optical structure. It uses a special optical path structure to divide the incident light into multiple beams. Each channel is placed with polarization devices with different polarization directions, and four polarization images with different degrees of polarization can be obtained on the detector [75]. The internal structure of the aperture real-time imaging system is precise. It can image polarization images of multiple polarization states in the target scene in real time, but the aperture division system has high requirements on lens and optical path structure, and the manufacturing difficulty is relatively high. In fact, it cannot completely achieve the perfect imaging of beam division and four quadrants, so it is relatively sensitive to noise, resulting in large error of polarization image [24].

In 2016, Fan et al. designed and implemented a new type of camera-based aperture bionic polarization navigation sensor and proposed a robust measurement method based on the least square method. Using this method, all outputs of the cell were measured and compared with the single Rayleigh scattering model. The sensor has better performance, 80% polarization angle error is less than 2°, and 40% deviation is less than 1° [76]. The structure of the sensor is shown in Figure 10d. Compared with other sensor structures, this structure can not only ensure that the incident light is parallel to the polarizer but also ensure that the polarization of the light remains unchanged when refracted in the lens. However, there is still a large space for improving the robustness and accuracy of the measurement. Especially for the polarization sensor based on multiple cameras, the response inconsistency error of each camera and the misalignment angle error of the polarizer are often ignored and uncalibrated. This will reduce the accuracy of polarization measurement [76].

In 2016, Zhang et al. proposed a new method based on the hand-held light field camera and radial polarizer [77]. This method can be classified as aperture coded light field capture method. As shown in Figure 11a, the schematic diagram of the imaging system is improved at the aperture plane of the handheld light field camera in this system, and the radial polarizer is used to replace the linear polarizer triad. The radial polarizer is composed of an S-wave plate (radial polarization converter), and linear polarizer is set in the main mirror. This imaging system can not only detect the polarization mode of the skylight in the whole sky at one time, but also has more robust performance. Figure 11b shows the photo of the actual sensor built as shown. The disadvantage of this device is that it is difficult to decompose the low frequency information of polarization mode and the high frequency part caused by microlens array, which affects the accuracy of polarization detection. It can further explore the feasibility of applying wavelet transform to improve the real-time performance of the system.

The aperture-type imaging sensor divides the scene light into multiple beams of polarized light waves through the aperture or polarization grating array and generates multiple polarized images in a single direction on an image sensor. There is spatial error in the generated image, and the image sensor also has the problem of inconsistent light response, which should be the next research direction. Although the aperture sensor meets the requirements of real time and convenience, it cannot adjust the field of view and resolution, and it has poor anti-interference ability and detection accuracy.

#### 4.2.4. Focal Plane Spectroscopic Sensor Based on Real-Time Imaging System

In recent years, with the rapid development of micro-nano manufacturing technology, the precision of products manufactured by nanotechnology is also getting higher and higher. With the continuous maturity of Ultraviolet Nanoimprint Lithography (UV-NIL) [78], Electron Beam Lithography (EBL), and Focused Ion Beam (FIB) technology [79], the detection, analysis, and processing of micro-nano patterns and micro-nano structures become easier. Therefore, the Division of Focal Plane (DoFP) focal plane splitting polarization sensor [78] came into being. The focal plane splitting imaging sensor is a polarization device processed on the focal plane of the detector [80]. This is a kind of polarization imaging sensor that can meet both real-time requirements and high accuracy and integration. The focal plane split polarization sensor can use a single image sensor to obtain polarization information in multiple directions at the same time. Such sensors can integrate the pixelated polarization array onto the image sensor to detect polarization information in different directions [78].

The Chu research group produced the following research: In 2015, Liu et al. proposed a polarization grating structure for nano-imprint grating, which can eliminate assembly errors. The angle measurement accuracy of the sensor in indoor and outdoor experiments is ±0.02° and ±1.3° [81]. In 2019, Ran Zhang et al. proposed a new DoFP polarimeter system, which uses a relay lens to separate the micro-polarizer array and image sensor [82]. This system can realize segmented local adaptive threshold segmentation and convolution interpolation calculation. The results show that the average error deviation of the degree of polarization after calibration is less than 1.5%, and the average error deviation of the polarization angle is less than 0.2%. The advantage of their camera system is that they can combine different micro-polarizer arrays, image sensors, and lenses to measure the multidimensional information of polarized light for different applications. In 2022, Liu et al. designed a bionic polarized light sensor based on the artificial compound eye [83]. As shown in Figure 12, the artificial compound eye is composed of a microlens array and diaphragm, and the integrated polarization detector is used to imprint the polarization grating onto the surface of CMOS by nano-imprint technology. The advantage of this sensor is that the diaphragm is introduced to avoid the crosstalk between lenses, and the nano-imprint process is adopted to reduce the manufacturing cost. The disadvantage is that the fabricated nanowires will collapse or be damaged, resulting in an inconsistent optical response of the polarizer. To solve this problem, Liu et al. used a multi-threshold segmentation algorithm to extract and classify effective pixels: they first defined the effective pixel value of the raster according to Marius’ law. If the raster or pixel fails, the light intensity change rule of the pixel will be different from the defined effective pixel value function change rule, and then determine whether the pixel is an effective pixel by calculating the variance between all pixels and effective pixels. After testing, the error is about 0.275°, and the standard deviation is about 0.115°. In 2022, Liu et al. used the proposed nano-imprint lithography process to produce a polarization sensor, and then integrated the image chip and nano-grating to produce a polarization chip. The polarization angle measurement error of the polarization navigation sensor manufactured was within 0.2° [84]. Figure 13a is the photo of the integrated polarization sensor, Figure 13b is the partial enlarged detail photo of the electrode, and Figure 13c is the Scanning Electronic Microscopy (SEM) image of the multidirectional nanograting.

The Hu research group produced the following research: In 2017, Han et al. proposed a pixelated polarized light compass [85]. This new polarization sensor is mainly composed of a CCD camera, a wide-angle lens, and a pixelated polarizer array. The sampling rate of this sensor can reach 80 frames per second. One image is enough to calculate the polarization mode of sky light, as shown in Figure 14a. The resulting pixel response is shown in Figure 14b. Han et al. proposed that when the incident light passes through the pixel polarizer array, the light intensity will be halved, and the pixel response of the CCD camera is also inconsistent. The inconsistency of this response depends on the internal characteristics and installation error of the camera [85]. They modeled the angle deviation and translation deviation, used the iterative least squares method to estimate the error parameters, and obtained the best results by minimizing the error square, The standard deviation of the orientation error of the sensor was 0.15°. In 2020, He et al. designed an array polarization sensor based on focal plane splitting. They used Kalman filtering to fuse the azimuth in the polarization information with the geomagnetic or micro-inertial measurement unit to obtain the optimal heading estimation [86]. Finally, in the vehicle experiment without satellite signal, the azimuth accuracy was better than 0.5°. In the next step, appropriate algorithms should be added to compensate the error between the grating and the pixel, and a new sensor with higher stability and smaller error should be developed to collect the polarized light information.

Other research teams have conducted research as follows: In 2010, Gruev et al. proposed an imaging sensor that integrates a polarization array made of aluminum nanowires with a CCD imaging array. Figure 15a is the block diagram of the CCD polarization imaging sensor. Each pixel is composed of a photodiode and two light-shielded buried channel CCDs. The polarization imaging sensor has a signal-to-noise ratio of 45 dB and captures the intensity, angle, and degree of linear polarization in the visible spectrum at a speed of 40 frames per second, with a power consumption of 300 mW. The aluminum nanowire polarization array with matching pixel spacing is covered on the CCD array of the photoelectric element. Figure 15b shows the SEM image of the aluminum nanowire with an angle of 45° [87]. In 2013, Sasagawa et al. proposed a polarization-sensitive pixel based on 65nm standard CMOS technology. They made fine metal patterns smaller than the wavelength of visible light using metal wire grids and designed and manufactured metal wire grid polarizers for image sensors on two pixels of 20 × 20 um; the extinction ratio of the polarizer at a wavelength of 750 nm was 19.7 dB [88]. In 2014, Zhang et al. designed a CMOS polarization image sensor and used numerical analysis to guide the design of the grid array, which is mainly used for the detection of living cells; the extinction ratio of the polarizer at a wavelength of 633 nm was 18.4 dB [89]. However, the nanoscale CMOS process used is not suitable for large-scale production and commercial manufacturing. In 2018, Garcia et al. proposed a polarization imaging sensor with high dynamic range [90], which can achieve a dynamic range of 140 dB and a maximum signal-to-noise ratio of 61 dB. The pixel of this sensor has a logarithmic response by operating a single photodiode in the forward bias mode. The pixelated polarization array is composed of aluminum nanowires with a height of 250 nm and a width of 75 nm. This kind of pixel circuit and pixelated polarization array can be integrated on a single chip to achieve a fast polarization imaging of 30 frames per second. In the same year, they also proposed an interpolation technique for the division of focal plane polarization image sensors, which can adaptively determine the polarization residual of pixels in four directions [91].

Most of the previous calibration models only considered the calibration method of the AOP. In 2020, Ren et al. analyzed the impact of the inconsistent extinction ratio of the orthogonal polarizer and the installation error on the sensor [92]. The installation error is shown in Figure 16. The extinction ratio coefficient is introduced into the polarization calibration model. This sensor uses a pixel-level polarization camera, and 2 × 2 matrix polarizers are used in front of every 2 × 2 photoelectric sensors. After calibration, the calculation accuracy of indoor AOP can reach 0.04°, and that of outdoor AOP can reach 0.71°. In 2020, Chen et al. used a pixel polarization camera to build a heading angle test device and installed the micro-polarizer on the photosensitive unit, so the field angle of view of the lens will not be reduced. After two heading angle calculation methods, the average errors of the heading angle were 0.332° and 2.304°, proving that the morning and evening are the best time periods for the application of bionic polarization navigation. However, this device has a great error in the presence of obstructions, which needs further study [24].

The focal plane beam-splitting sensor is realized by integrating the focal plane array and the micro-polarizer array, and the instantaneous field of view error is the inherent problem of this sensor. In 2020, Hao et al. proposed a method for the structural design of micro-polarizer arrays. They reduced the impact of Instantaneous Field of View (IFoV) by optimizing the configuration of micro-polarizer arrays [93]. By analyzing the spectral characteristics, frequency components, and crosstalk between the filtered images of micro-polarizer arrays, they converted the design of micro-polarizer arrays into a multi-objective optimization problem, and finally optimized their spectral distribution. A micro-polarizer array with better performance was designed. In 2022, Nie et al. proposed a metal gate substrate structure applied to the micro-polarizer array, which can effectively reduce optical crosstalk and improve the extinction ratio of the micro-polarizer array. The calculation results show that the extinction ratio of the structure they proposed is more than 10 times higher than that of the traditional micro-polarizer structure [94]. They proposed a micro-polarizer array with a metal mesh substrate, as shown in Figure 17a, showing the structure diagram of the micro-polarizer array. Figure 17b is the schematic diagram and characteristic parameters of the top sub-wavelength Al grating array, Figure 17c is the schematic diagram of the bottom substrate structure and characteristic parameters, and Figure 17d is the simulation schematic diagram of the single pixel structure. The metal structure in this substrate can eliminate the optical crosstalk caused by the diffraction effect. In the case of oblique incident light, the metal grating in this substrate can completely eliminate the possibility of oblique light entering adjacent pixels.

The focal plane polarization sensor uses a single image sensor to obtain polarization information in multiple directions at the same time. It integrates a pixelated polarization array on the image sensor, which can avoid optical crosstalk. The monolithic integration of micro-polarized array and the CMOS imaging sensor is a hot research topic at present. These sensors are called focal plane spectroscopic sensors [86]. The main task of manufacturing this kind of sensor is to design the micro-polarization array, and the design of this micro-polarization array mainly considers the following key issues. First, consider that the pixel spacing of the micro-polarization array must match the pixel spacing of the photodetector array. Secondly, the designed micro-polarization array must have a good extinction ratio to effectively extract polarization information. Thirdly, the crosstalk between adjacent pixels must be minimized. If the polarization information between adjacent pixels is mixed with each other, it will cause large errors. At the same time, there are methods for manufacturing polarization arrays using birefringent materials [95] and thin film materials [96]. However, small imaging array size, large pixel size, and small extinction ratio limit the use of these micro-polarization arrays. See Table 4 for the overview of sensors applied to focal plane spectroscopic type and based on micro-nano processing technology.

### 4.3. Polarized Light Sensor Based on Micro-Nano Processing Technology

In recent years, with the rapid development of micro-nano machining technology and precision machining technology, the manufacturing technology of polarization grating in nano-manufacturing technology has changed greatly. Micro-nano manufacturing technology makes it possible to manufacture micro-polarization detectors in the visible spectrum [96,117,118,119,120]. At present, polarized light sensors are developing in the direction of miniaturization, integration, diversification, high accuracy, and high robustness. A new generation of polarized light navigation measurement system based on multi-spectral measurement, multi-sensor fusion, and image-based intelligent algorithm technology will be more widely used [25]. In the middle of the 19th century, Rand et al. heated the PVA sheet and stretched it in one direction, so that the polymer molecules were arranged in a long chain in the stretching direction, and successfully commercialized the polarizer [121]. In the middle and late 20th century, the development of micro-nano technology promoted the emergence of multi-directional sensitive polarizers. In 1999, Nodin et al. made a subwavelength metal grating polarizer using silicon as the substrate by exposure etching and other methods, but the cost is expensive, and the process is complex, which is not conducive to commercialization [122]. In 2006, Momeni et al. used the idea of crystal birefringence to deposit opaque metal on the top of the metal to make a micro-polarization array to prevent the mixing of optical components with orthogonal polarization [110]. In 2010, Sarkar et al. proposed a real-time CMOS image sensor using a linear grid polarizer, which was made using 180 nm CMOS technology; the linear fitting error of the linear polarization angle is 6 %, and the correlation coefficient between the theoretical and experimental values is 0.985 [111]. Each pixel on the top of the photodiode has an embedded metal wire grid micro-polarizer, which is realized by the first metal layer of CMOS technology. As shown in Figure 18a,b, the profile of the wire grid polarizer and the sensor area map with different polarization angles are shown.

In 2011, Gruev et al. proposed an optimized method for fabricating micro-polarized arrays by interference lithography. The resulting array is composed of aluminum nanowires with a spacing of 140 nm, a height of 140 nm, and a width of 70 nm; the maximum extinction ratio of the filter array at 700 nm is 95 [112]. In 2012, Chu et al. made six double-layer metal grating polarizers with different directions on polycarbonate substrate using nano-imprint technology. The obtained polarizers can be used to eliminate the installation errors caused by discrete devices; when incident at a wavelength of 510 nm, the transmittance of TM polarized light is higher than 71%, and the extinction ratio is greater than 2100, with good application prospects [123]. In 2013, Gao et al. introduced a method for manufacturing aluminum metal nanowire grid polarizers using EBL and RIE. Among them, EBL technology provides a large degree of freedom to make different gratings in a limited small area, and RIE technology provides high-precision etching, which better preserves the design features [113]. In 2014, Chu et al. used nano-imprint lithography technology to integrate a double-layer nanowire polarizer with a photo-detector to form a polarized light sensor; the angle output error of this sensor is within ±0.1° [124]. Figure 19h is an optical picture of the integrated polarized light sensor. Figure 19g is the amplified optical image of the integrated polarized light sensor, and Figure 19a–f is the grating SEM image made based on the nano-imprint lithography process. They measured the output voltage signal of the polarization sensor, as shown in Figure 20a. They pointed out that the reason for the different amplitude is the defect caused by NIL process, which leads to the different performance of polarization sensor [124]. To solve this problem, they normalized the output of the polarization sensor and modeled and compensated the angle output error by polynomial curve fitting.

For underwater polarization mode, in 2018, B. Powell et al. proposed a visible spectrum imaging polarimeter based on underwater polarization mode, as shown in Figure 20b, which can determine the geographical location [125]. The sensor simulates a polarization-sensitive vision system by integrating a polarizer in each pixel of the camera. The polarizer is aligned by parallel aluminum nanowires and deposited on each pixel of the image sensor. They used a general optimization algorithm to match the measured polarization angle with the prediction of the basic single scattering model of underwater light to infer the position of the sun, and then used the k-nearest neighbor algorithm to process the residual in the model. Finally, they reduced the root mean square errors of the sun’s polarization angle and elevation angle to 6.02° and 2.92°, respectively. In 2022, Hu et al. developed a solar positioning algorithm based on the underwater refraction polarization mode in the Snell window [126]. They used Snell’s law and Fresnel’s refraction formula to decouple the bending and polarization deflection of re-fracted light and obtained the celestial polarization map based on underwater measurement, and then used the degree of polarization as the weight factor for calculating the E vector. The results show that the root mean square errors of the solar zenith and azimuth were 0.3° and 1.3°, respectively. The underwater imaging polarization sensor is shown in the Figure 20e. The limitation of the refraction polarization algorithm proposed by them is that it is only applicable to flat water. Future work can optimize and improve the algorithm to further expand the application range. For underwater polarization navigation, the adaptation of the wave surface and the depth of application should also be quantified. In 2020, Guan et al. constructed a multi-orientation nanowire polarizer using nanoimprint lithography technology. The sensor prototype is shown in Figure 20c. The sensor is a polarization sensor with high accuracy and strong robustness [114]. They proposed an adaptive algorithm based on the information entropy theory, which can eliminate the influence of random obstacles in the sensor field under changing light conditions. Finally, the error of AOP can be controlled within ±0.1°, and the angle fluctuates between ±0.4°. This sensor should be further optimized in remote navigation, and an accurate compensation model should be established according to time and position to better eliminate noise. In 2022, Guan et al. built a real-time polarization sensor based on ultraviolet nano-imprint lithography (UV-NIL). As shown in Figure 20d, UV-NIL integrates the micro-polarizer array onto the packaged CMOS sensor without changing the CMOS structure, and uses FPGA as the processing unit to demodulate, reconstruct, and display the polarization image. In order to reduce the error and improve the reliability of the sensor, a calibration method based on discrete Fourier transform and relative entropy was also proposed [78]. They first considered the edge vulnerability of CMOS chips and proposed a flexible UV-NIL for the unformed area protection integration process. It provides a cost-effective and compatible method with the assembly process of standard semiconductors for polarization image sensors. The disadvantage is that in the process of UV-NIL, the collapse of nanowires leads to inconsistent optical response of polarizers in different directions. Future work can be devoted to improving the surface profile of micro-polarizer arrays.

This kind of polarization camera is usually manufactured by manufacturing detector array and micro-polarizer array, respectively, and then integrated by aligning with each other. Therefore, when the incident light angle is non-zero, there is a small vertical error between the polarizer layer and the detection layer, which will lead to the crosstalk problem of adjacent pixels. In addition, micro-polarizers manufactured with linear grid patterns are difficult in terms of achieving the polarization purity that can be achieved by a single polarizer. In response to this came the micro-grid polarization camera [127]. After comparing the problem, Hagen et al. proposed a method to calibrate average value and variance of attenuation and orientation characteristics; the accuracy of calibration pixel polarization characteristics under Gaussian and Poisson noise conditions was given, and the statistical instability of the extinction ratio as a parameter was proven. In their calibration and measurement work, the inter-pixel variation of polarization characteristics was estimated. The way in which to eliminate the time error was also described.

In 2022, Sasagawa et al. proposed an image sensor with high sensitivity to detect polarization change. They built an optical system with a two-layer structure, including external polarizers and polarizers on the pixel array. By arranging two polarizers under the best conditions, they were able to achieve high polarization change detection performance. In addition, they also proved the applicability of electric field distribution imaging using electro-optical crystals for the measurement of weak polarization change distribution [115]. In 2022, Zheng et al. designed a polarization image sensor. The sensor uses the CMOS standard process to manufacture the interconnection metal grid structure. The proposed metal grid structure has dual operation modes, namely, the polarization image sensor mode of the visible spectrum and the physical non-cloning function mode of the near-infrared spectrum. The manufacturing process adopts 65 nm, 1P8M standard CMOS technology, and the grating period is designed to be greater than 200 nm and less than 400 nm. Finally, the detection range of the designed polarization image sensor is 400 nm–900 nm [118].

## 5. Conclusions and Outlook

After hundreds of millions of years of evolution, organisms in the natural environment have evolved various abilities to perceive environmental information, including the ability of some insects and birds to use polarized light in the sky for navigation. The polarized light navigation method developed by imitating the sensitive mechanism of insect compound eyes to polarized light is a new method that has just emerged in recent years. The research and development of a bionic polarized light navigation sensor has become a hotspot in the field of bionic and navigation research. In this paper, a systematic and in-depth review of polarized light navigation sensors is carried out. Seven types of imaging sensors are reviewed and compared in detail, including basic point source polarized light sensors, point source polarized light sensors applied to integrated navigation, sensors based on a time-sharing imaging system, amplitude-splitting type based on a real-time imaging system, aperture splitting type, focal plane splitting type, and imaging sensors based on micro-nano processing technology. The research progress of typical sensors in the past two decades is described.

Compared to traditional methods that require expensive high-definition cameras, basic point source polarized light sensors can provide excellent results at extremely low costs. However, basic point source sensors also have many problems, such as the following: (1) Solar blurriness: blurriness between 0° and 180°, 180°, and 360° on the edges of the sensor pixels produced. (2) Spectral sensitivity issue: due to certain limitations in the range of the spectrum. (3) Circuit noise issue: due to the presence of certain circuit noise in electronic circuits, the accuracy of the sensor may be altered. Future research work needs to focus on the following aspects: (1) When testing sensor accuracy, factors such as different UV indices and uncontrolled lighting conditions should be considered. (2) Improve sensor navigation performance in terms of spectral sensitivity. (3) Due to circuit noise issues that can reduce sensor accuracy, the next step is to study methods to maintain high accuracy when polarization is large and small. Due to the inevitability of electronic circuit noise, researchers have proposed integrating more navigation sensors into integrated navigation to further improve navigation accuracy. The point source sensors used in integrated navigation mainly have the problem of correcting installation errors between polarization sensors and other navigation sensors. Future research work should focus on developing information fusion, optimizing navigation control algorithms, and exploring higher integration and smaller multi-sensor systems. In addition, a visual position recognition framework with multiple spatial scales should be expanded and optimized in an open large-scale environment, aiming to build a complete biologically inspired system with machine-learning-based fusion solutions for map drawing, navigation, and path planning. The main feature of time-sharing imaging sensors is that images with multiple polarization angles are captured in different time periods and cannot simultaneously capture polarization images with multiple polarization angles. This inevitably leads to time-sharing errors. In addition, due to the need to rotate the polarizer in time-sharing imaging, there are also certain installation and rotation errors. Future work should focus on addressing error issues. Due to the complexity of the optical path structure, amplitude divided sensors are expensive and not conducive to commercial production. Future work should focus on the development of simple and inexpensive multi optical path structures. The aperture splitting sensor can image multiple polarization images of the target scene in real time, but the response inconsistency error of each camera and the misalignment angle error of the polarization filter are often ignored and uncalibrated, which will significantly reduce the accuracy of polarimetry. Future work should improve the measurement accuracy and timeliness of polarization image processing. In addition, polarization mode measurement in a wide field of view (FOV) should also be developed, as a wider field of view can help improve the accuracy of sensors, anti-interference ability, and reliability. The focal plane spectroscopic sensor and polarized light sensor based on micro nano processing technology have advantages such as small size, good stability, wide detection range, high angular frequency, and real-time detection, which have broad application prospects. However, these two types of sensors also have certain drawbacks: due to the gap between the nanowire polarizer and the image plane in the glass packaging cover of CMOS sensors, the integration level needs further improvement. Future work should focus on establishing accurate compensation models based on time and location. In addition, in order to cope with the high-frequency noise caused by the carrier movement of polarization sensors, a more effective heading measurement filtering algorithm should be established. In addition to high-frequency noise, polarizers made by micro nano processing technology should also further increase the signal-to-noise ratio of pixels due to the influence of photon shot noise and other noises.

Although the current bionic polarized light navigation sensor has made some research progress, it still faces many challenges and needs further study in the following aspects: First, the polarization angle detection experiments are mostly carried out under the conditions of clear and cloudless sky. When the sky is cloudy, the polarization angle measurement error is large. In the follow-up research, the algorithm should be further optimized to eliminate noise and realize the correct calculation of the polarization angle in extreme weather. Secondly, the indoor and outdoor data collected in the experiment are inconsistent in error, so it is necessary to further study the visual nervous system of organisms that rely on polarized light for navigation, improve the sensor structure, and improve its performance. Third, at present, most of the sensor platforms have low integration, and all components and modules are discrete. In the future, integrating sensors should be considered to reduce the volume. Therefore, the current research field of the polarized light navigation sensor mainly takes miniaturization and integration as the development trend. It is of great research significance and practical application value to study a new type of polarized light navigation sensor that is completely autonomous and has the advantages of compact structure, good real-time performance, and low cost. It can provide an effective means of navigation for various social fields such as transportation, scientific research, and resource exploration, and thus produce significant economic and social benefits [25].

## Figures and Tables

**Figure 1 sensors-23-05848-f001:**
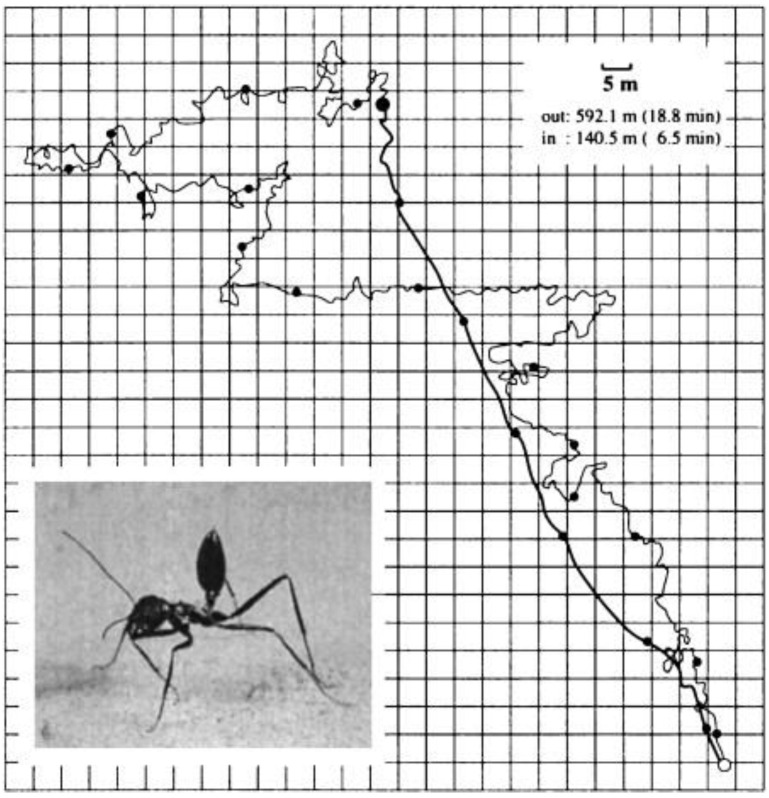
Navigation trajectories of sand ants [3].

**Figure 2 sensors-23-05848-f002:**
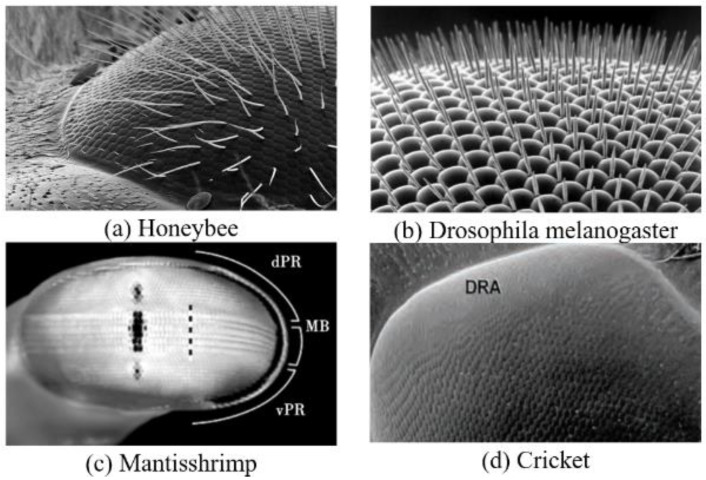
Compound eye morphology of various insects. (**a**) The structure of bee’s compound eye and the arrangement of Microvillus. (**b**) Compound eye structure and Microvillus arrangement of Drosophila melanogaster. (**c**) Cross section structure of the compound eye of Mantisshrimp. dPR: dorsal Peripheral Region. MB: Mid-Band region. vPR: ventral Peripheral Region. (**d**) The compound eye DRA region of crickets.

**Figure 3 sensors-23-05848-f003:**
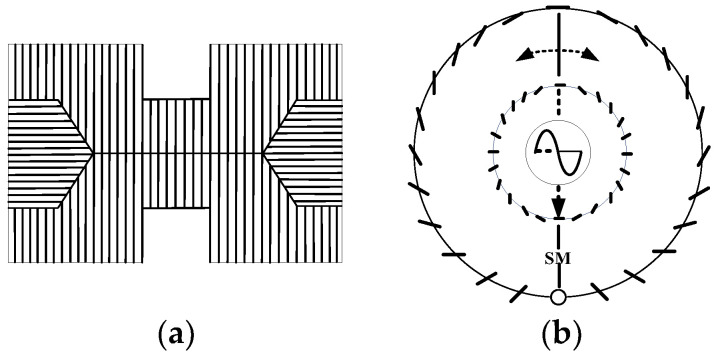
(**a**) Schematic diagram of the structure of biological neural rods. (**b**) Schematic diagram of the arrangement of biological neural rods. SM: Direction of the solar meridian.

**Figure 4 sensors-23-05848-f004:**
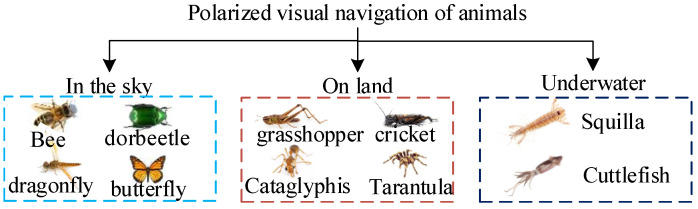
Polarized visual navigation of animals.

**Figure 5 sensors-23-05848-f005:**
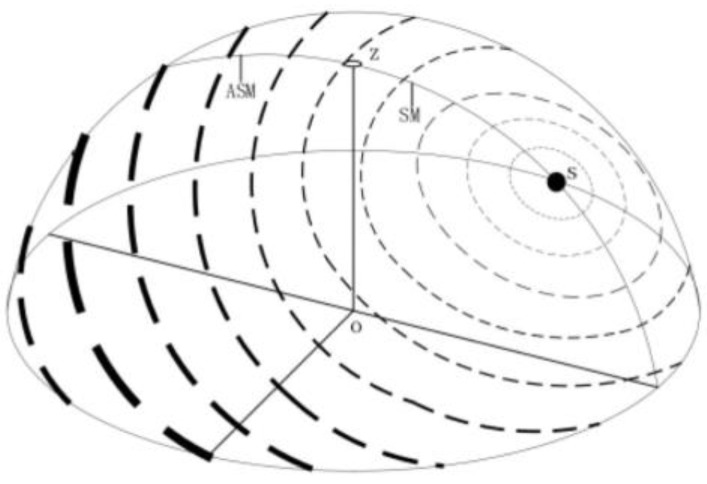
Polarization distribution patterns of the atmosphere in the sky. SM: solar meridian. Asm: anti-solar meridian.

**Figure 6 sensors-23-05848-f006:**
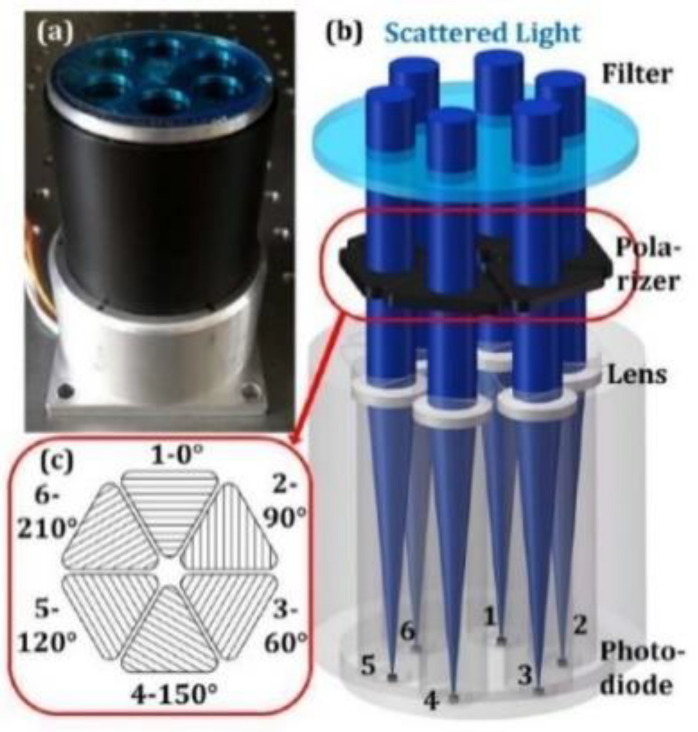
(**a**) Photo of bionic polarization sensor [34]. (**b**) Scattering light path [34]. (**c**) Direction of six-polarizers [34].

**Figure 7 sensors-23-05848-f007:**
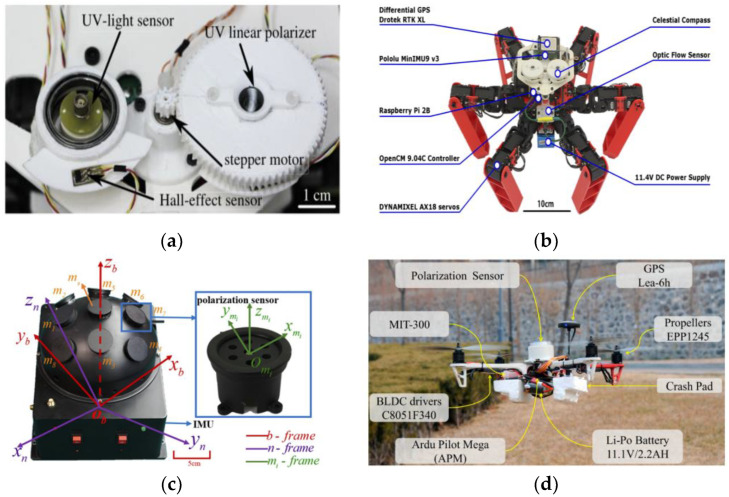
(**a**) Detection device of polarized light in ultraviolet band [35]. (**b**) Structural drawing of the hexapod robot [38]. (**c**) Bionic attitude heading reference system [40]. (**d**) Photos of the quadrotor UAV with a polarized light sensor [45].

**Figure 8 sensors-23-05848-f008:**
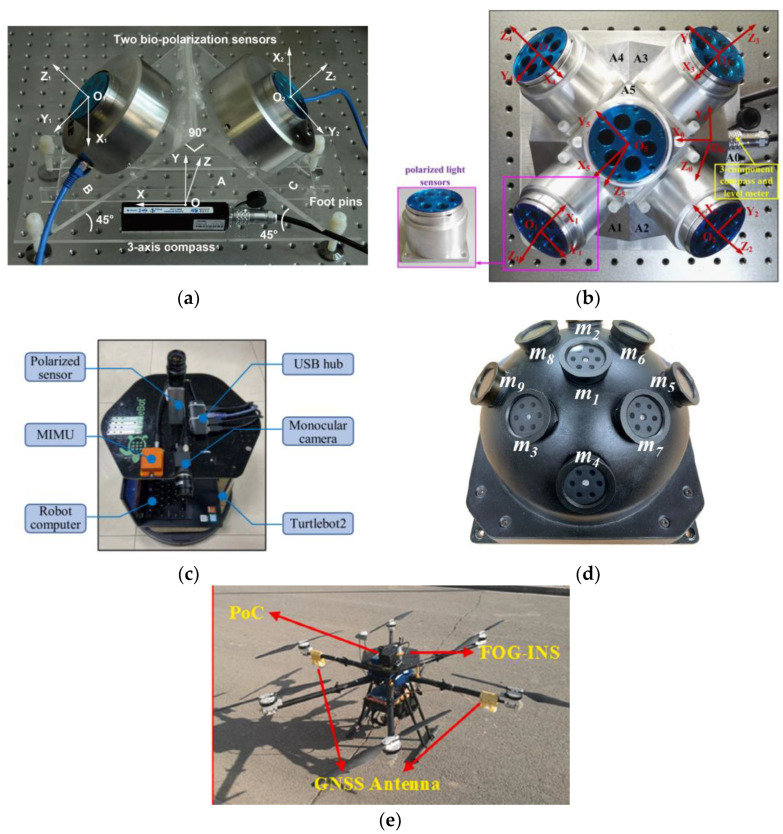
(**a**) Two E vector sensor measuring devices [46]. (**b**) Five E vector sensor measuring devices [47]. (**c**) Bionic multi-sensor navigation control system [49]. MIMU: Miniature Inertial Measurement Unit. (**d**) Polarized light sensor unit [54]. (**e**) Six rotor UAV with polarized light sensor [55]. PoC: Polarization Compass. FOG-INS: Fiber Optic Gyroscope Inertial Navigation System. GNSS: Global Navigation Satellite System.

**Figure 9 sensors-23-05848-f009:**
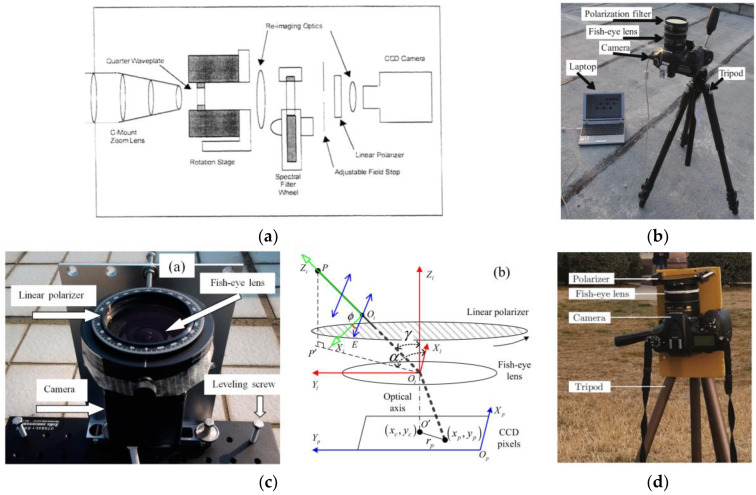
(**a**) System schematic diagram of multispectral imaging polarizer [58]. (**b**) Polarization imaging system [60]. (**c**) Polarized light sensor system [61]. (**d**) Polarization imaging system [62].

**Figure 10 sensors-23-05848-f010:**
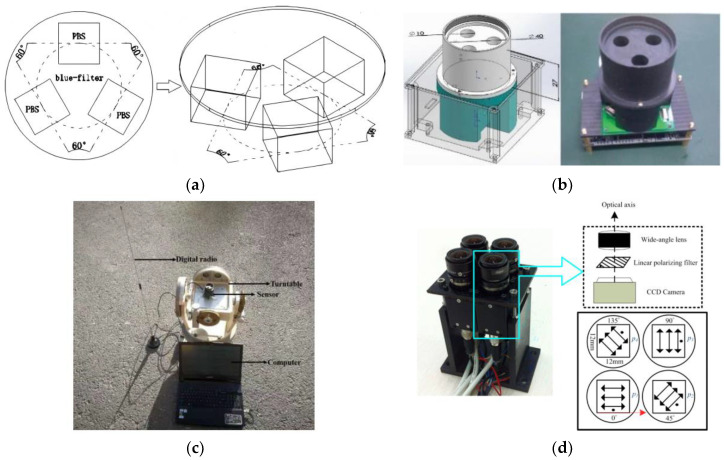
(**a**) The constructed polarization sensor based on the polarization beam splitter has a common filter at the top and three polarization beam splitters at the bottom [72]. PBS: Polarizing Beam Splitter. (**b**) Optical photo and structure of the traction tube [73]. (**c**) Polarized light sensor for measuring the error model [74]. (**d**) Sensor photos [76].

**Figure 11 sensors-23-05848-f011:**
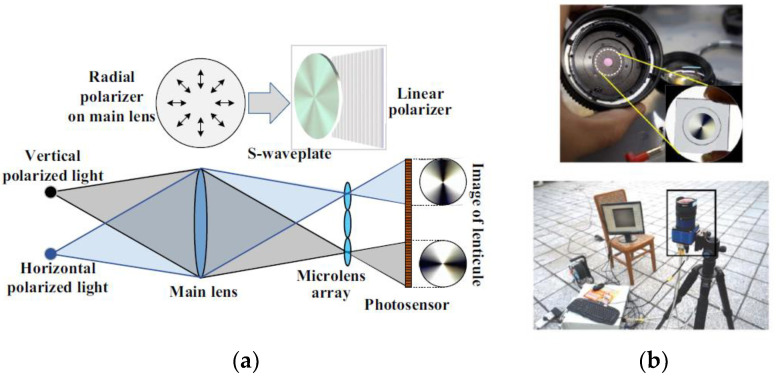
(**a**) Schematic diagram of the imaging system [77]. (**b**) Photo of the actual polarized light sensor [77].

**Figure 12 sensors-23-05848-f012:**
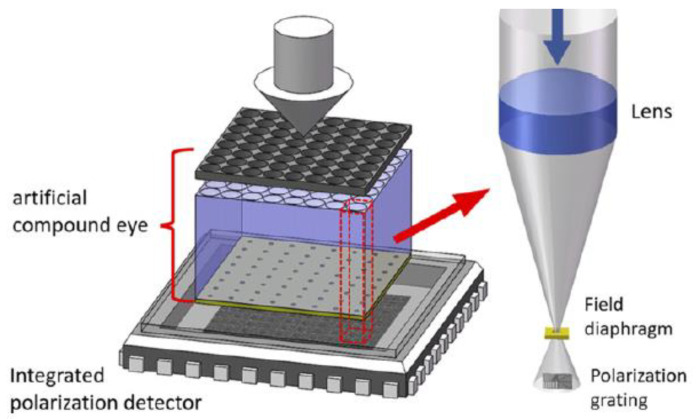
Schematic diagram of polarized light sensor and measuring optical path [83].

**Figure 13 sensors-23-05848-f013:**
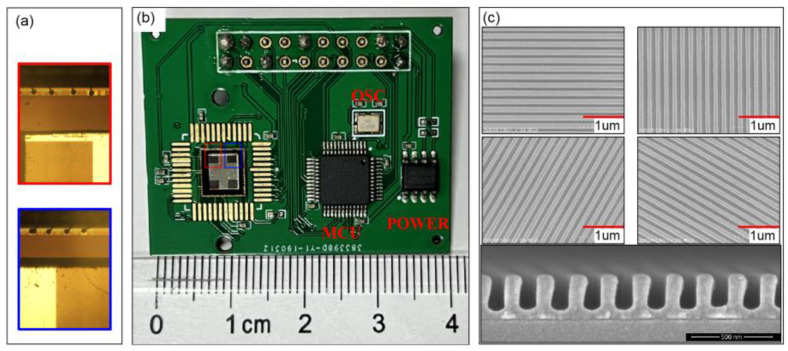
Schematic diagram of an integrated polarized light sensor [84]. (**a**) The photograph of the integrated polarization sensor including the polarization chip, MCU (Microcontroller Unit) and Power [84]. (**b**) The photographs of partially enlarged details of nanogratings and adjacent electrodes of the polarization chip (red and blue boxes) in (a) [84]. (**c**) SEM (Scanning Electron Microscope) images of multidirectional nanogratings on the test piece of silicon by the same integration process [84].

**Figure 14 sensors-23-05848-f014:**
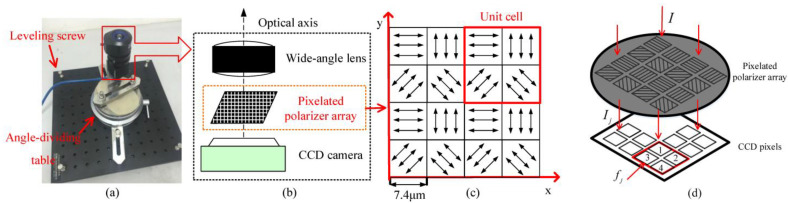
(**a**) the pixelated polarized light compass [85]. (**b**) the installing structure of the sensor [85]. (**c**) layout design of the pixelated polarizers. [85]. (**d**) Schematic representation of the response of CCD pixels [85].

**Figure 15 sensors-23-05848-f015:**
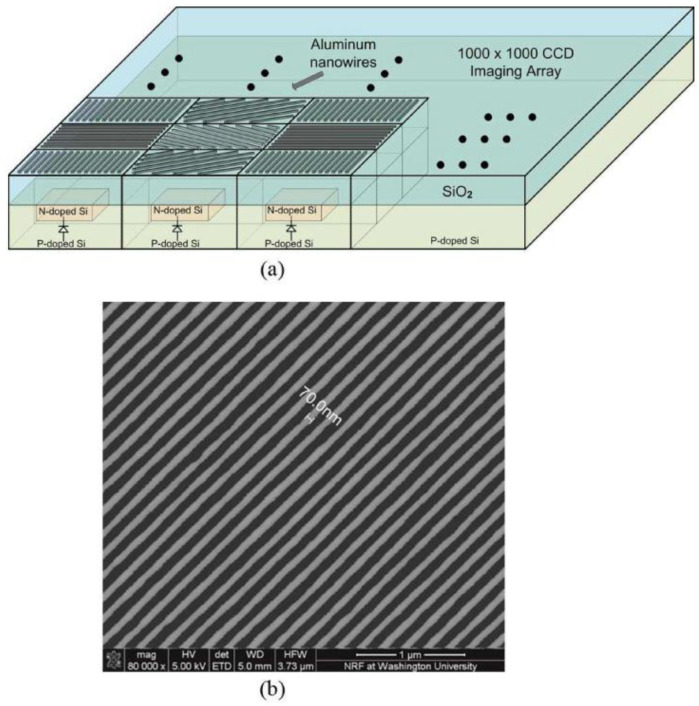
Block diagram of a CCD polarized light sensor [87]. (**a**) Block diagram of the integrated CCD polarization image sensor. The imaging array is covered with a pixel pitch-matched micropolarization filter array with four different orientations offset by 45°. The micropolarization filter array is composed of aluminum nanowires with 70 nm width, 140 nm pitch and 70 nm height [87]. (**b**) SEM image of the aluminum nanowire polarization filters oriented at 45 degrees [87].

**Figure 16 sensors-23-05848-f016:**
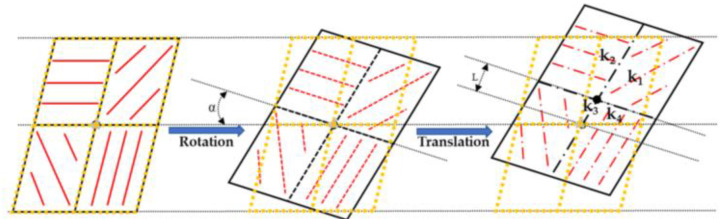
Installation error of the polarization unit [92].

**Figure 17 sensors-23-05848-f017:**
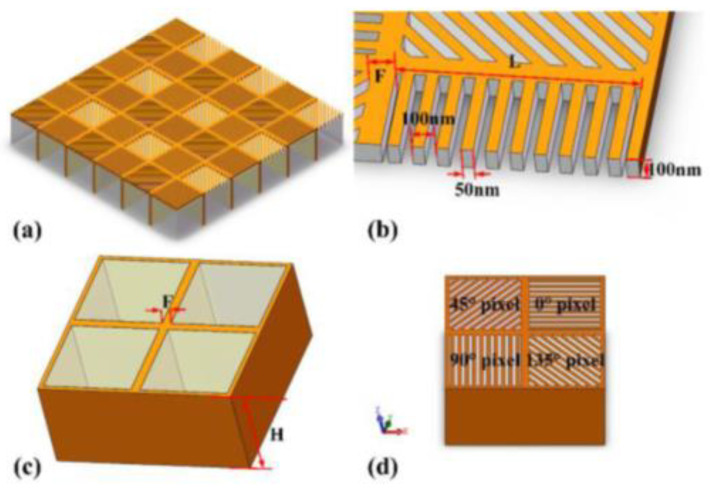
Schematic diagram of the micro-polarizer array [94]. (**a**) Schematic diagram of the MPA (Micro-Polarizer Array) structure. The yellow part represents the metal material, and the translucent gray part represents SiO2 [94]. (**b**) Schematic diagram and characteristic parameters of the subwavelength Al grating array at the top of (a) [94]. (**c**) Schematic diagram structure and characteristic parameters of the substrate at the bottom of (a) [94]. (**d**) Schematic diagram of the simulation of a single super pixel structure [94].

**Figure 18 sensors-23-05848-f018:**
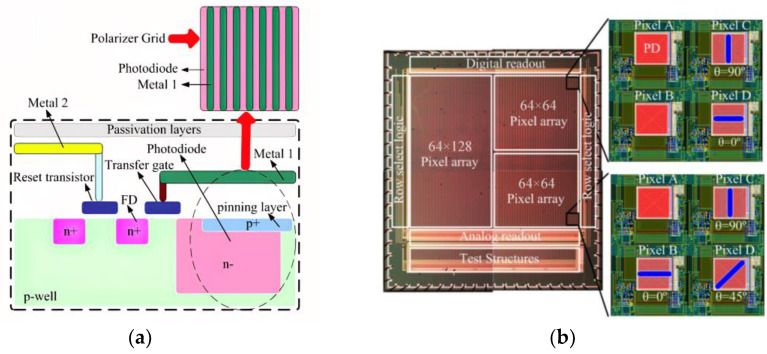
(**a**) Profile of the grid polarizer [111]. (**b**) Map of the sensor area with different polarization angles [111].

**Figure 19 sensors-23-05848-f019:**
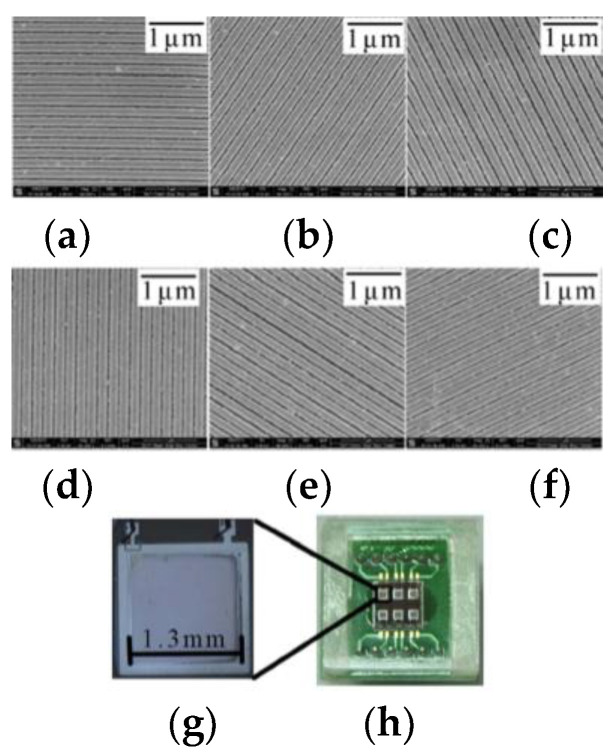
Integrated polarization sensor optical image [124]. (**a**)–(**f**): the SEM images of the grating in the sensitive area. The orientations are, correspondingly, 0°, 60°, 120°, 90°, 150° and 30° [124]. (**g**) enlarged optical image of the PDP (Polarization Dependent Photodetector) [124]. (**h**) optical image of the PDPs in one chip [124].

**Figure 20 sensors-23-05848-f020:**
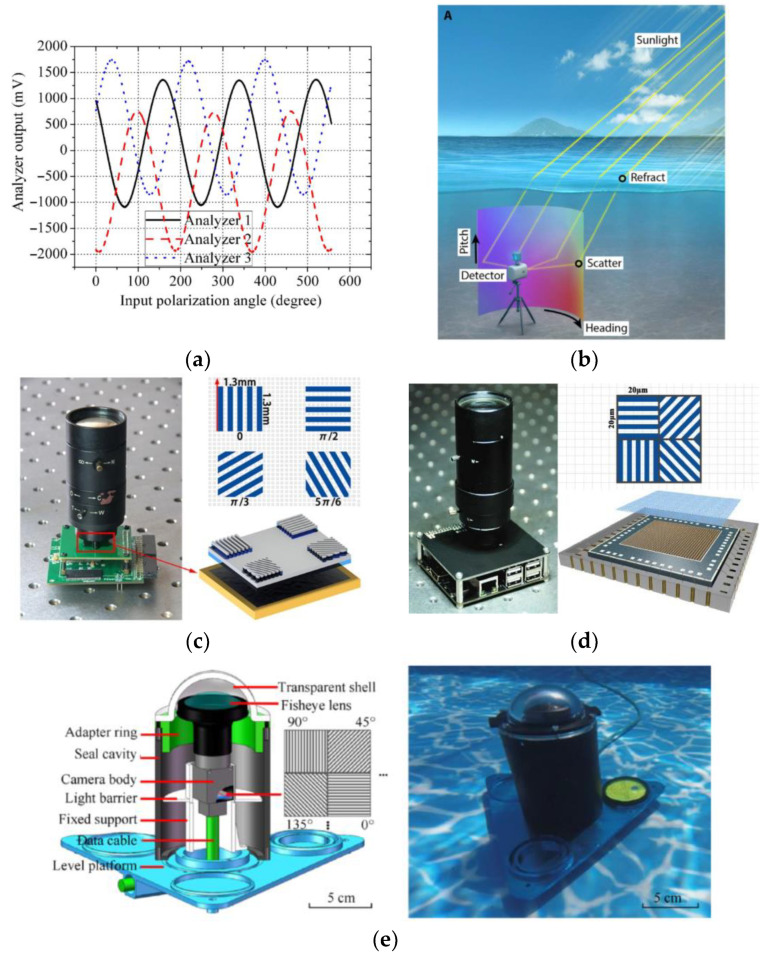
(**a**) Polarization sensor output voltage [124]. (**b**) Underwater polarization model diagram [125]. (**c**) Vibration sensor prototype diagram and polarizer distribution [114]. (**d**) Vibration sensor prototype diagram and polarizer distribution [78]. (**e**) Underwater polarization sensor. In the picture “⋯” meaning is that the camera pixels are arranged in an array according to this polarization angle. [126].

**Table 1 sensors-23-05848-t001:** Overview of various sensors.

Sensor Classification		System Composition	Typical Sensor	Reference	Advantages	Disadvantages
Point source polarized light sensor	Basic point source sensor	Photodiode, polarizer, filter	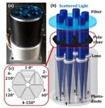	[3,4,5,18,31,32,33,34,35,36,37]	Simple structure, small volume, and cost.	It is easy to be disturbed by the external environment and has poor stability.
	Point source sensor for integrated navigation	Polarized light navigation, micro inertial navigation, Global Positioning System (GPS) navigation	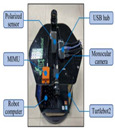	[38,39,40,41,42,43,44,45,46,47,48,49,50,51,52,53,54,55,56]	Improve navigation performance and accuracy, and overcome the disadvantages of single sensor.	High production cost and large volume.
Imaging polarized light sensor	Time-sharing imaging	Camera, turntable, polarizer	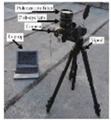	[57,58,59,60,61,62,63,64,65,66,67,68,69,70]	Simple production and low cost.	The orientation error and assembly error are large.
	Fractional amplitude imaging	Camera, polarization beam splitter, delayer	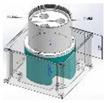	[71,72,73,74]	Multidirectional polarization angle image detection is available.	Due to the large volume and different imaging brightness of different detectors, the orthogonal error of polarizers is inevitable.
	Aperture splitting	Light field camera is usually used	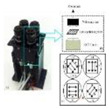	[75,76,77]	It can image polarization images of multiple polarization states of the target scene in real time.	The requirements for lens and optical structure are high, and the relative sensitivity of noise leads to large polarization image error.
	Focal plane light splitting	Charge coupled device (CCD) camera, complementary metal oxide semiconductor (CMOS) camera, pixelated polarization array	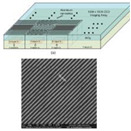	[78,79,80,81,82,83,84,85,86,87,88,89,90,91,92,93,94,95,96]	Good real-time performance, high precision, and high integration.	High fabrication cost and nanowire error lead to inconsistent optical response of polarizers in different directions.

**Table 2 sensors-23-05848-t002:** Overview of point source polarized light sensor.

Authors	Years	Reference	Sensor Type	Navigation Mode	Whether It Can Work in the Electromagnetic Interference Environment	Spectrum Detection Range	Technical Solutions	Main Technologies/Algorithms/Contributions
Dimitrios Lambrinos et al.	1997	[2]	Basic point source polarized light sensor	Polarized light navigation	Yes	Blue light	Photodiode	Scheme of polarized light navigation
	2000	[3]				Blue light		Path integration and visual navigation mechanism
Jinkui Chu et al.	2008	[4]				Blue light		A new polarized light sensor
Evripidis Gkanias et al.	2019	[32]				Ultraviolet		Correction of sensor array tilt algorithm
Javaan Chahl et al.	2012	[31]				Green light and ultraviolet light		Polarized light compass and its calibration method
Kaichun Zhao et al.	2009	[33]				400 nm–520 nm		Angle output algorithm
Yinlong Wang et al.	2019	[34]				320 nm–730 nm		Centrally symmetric algorithm, discontinuous algorithm
Julien Dupeyroux et al.	2019	[35]				270 nm–375 nm		Two-pixel polarized light sensor
	2019	[18]				270 nm–375 nm		Waterproof ultraviolet polarized light sensor

**Table 3 sensors-23-05848-t003:** Overview of point source sensor for integrated navigation.

Authors	Years	Reference	Sensor Type	Navigation Mode	Whether It Can Work in the Electromagnetic Interference Environment	Spectrum Detection Range	Technical Solutions	Main Technologies/Algorithms/Contributions
Dupeyroux et al.	2019	[38]	Point source sensor for integrated navigation	Polarized navigation/GPS	No	270 nm–375 nm	Photodiode	Path integral integrated navigation strategy
Du et al.	2020	[39]		Polarized navigation/SINS	Yes	400 nm–700 nm	Ambient light sensor	Static autonomous initial alignment algorithm
Liu et al.	2021	[40]		Polarized navigation/MIMU	Yes	400 nm–700 nm	Ambient light sensor	Bionic attitude and heading reference system
Chu et al.	2009	[44]		Polarized navigation/GPS	No	Blue light	Photodiode	Design of fuzzy logic controller
Chu et al.	2018	[45]		Polarized navigation/MIMU/GPS	Yes	Blue light	Photodiode	Attitude determination system assisted by polarized light sensor
Wang et al.	2015	[46]		Polarized light navigation/geomagnetic navigation	No	Blue light	Photodiode	Combined positioning of polarized light and geomagnetic field
Wang et al.	2018	[47]		Polarized light navigation/gravitational field/geomagnetic field	No	Blue light	Photodiode	Orthogonal vector algorithm
Zho et al.	2021	[49]		Polarized navigation/MIMU	Yes	400 nm–700 nm	Integrated polarization sensor	Sensor joint calibration algorithm and adaptive integration algorithm
Xie et al.	2021	[50]		Polarized light navigation/MIMU/magnetometer	No	400 nm–700 nm	Polarization based camera	Dead reckoning algorithm
Fan et al.	2022	[51]		Polarized navigation/MIMU	Yes	400 nm–700 nm	Polarization based camera	Optimal orientation algorithm and two-dimensional visual position recognition technology
Fan et al.	2022	[52]		Polarized navigation/INS	Yes	400 nm–700 nm	Polarization based camera	Robust bionic polarization skylight orientation algorithm and polarization mode consistency algorithm
Hu et al.	2021	[54]		Polarized navigation/MIMU	Yes	400 nm–700 nm	Ambient light sensor	Adaptive partial feedback algorithm for attitude angle
Zhao et al.	2022	[55]		Polarized navigation/MIMU/GNSS	No	400 nm–700 nm	Polarization based camera	Heading error modeling and compensation algorithm for attitude change of polarization compass

**Table 4 sensors-23-05848-t004:** Overview of point source sensors for integrated navigation.

Authors	Years	Reference	Manufacturing Technology	Technical Solutions	Main Technologies/Algorithms/Contributions
Liu et al.	2015	[81]	Nanoimprinting technology	Integrated polarization sensor	Applying statistical theory to interval division algorithm of polarization angle
Zhang et al.	2019	[82]	Nanoimprinting technology, inductively coupled plasma etching (ICP), thermal evaporation process	Camera-based	Segmental local adaptive threshold segmentation and convolution interpolation algorithm
Liu et al.	2022	[83]	Flexible nano-imprint technology	Integrated polarization sensor	Multi-threshold segmentation algorithm
Ze Liu et al.	2022	[84]	Nanoimprint lithography process	Integrated polarization sensor	Combining traditional optical imprinting technology with nanoimprinting technology
Gruev et al.	2010	[87]	Electron beam evaporation deposition technology, reactive ion etching (RIE) technology	Integrated polarization sensor	Monolithic integration of polarization array and CCD imaging array
Sasagawa et al.	2013	[88]	65 nm standard CMOS technology	Integrated polarization sensor	Using deep submicron CMOS technology to design fine metal patterns smaller than visible light wavelength
Zhang et al.	2014	[89]	65 nm standard CMOS technology	Integrated polarization sensor	The design of grid array is guided by numerical analysis
Garcia et al.	2018	[90]	180 nm standard CMOS technology	Integrated polarization sensor	A polarization imaging sensor with high dynamic range is proposed
Momeni et al.	2006	[110]	1.5 micron double-n-well CMOS process, lithography and deposition technology	Integrated polarization sensor	Install the birefringent micro-polarizer on the CMOS chip
Sarkar et al.	2010	[111]	180 nm standard CMOS technology	Integrated polarization sensor	Polarization sensor with high integration
Gruev et al.	2011	[112]	Interference lithography process and micromachining process	Integrated polarization sensor	Fabrication of micro-polarization array by interference lithography and micromachining
Shengkui Gao et al.	2013	[113]	Electron beam lithography (EBL), reactive ion etching (RIE)	Integrated polarization sensor	Preparation of aluminum pixelated polarization structure by electron beam lithography (EBL) and reactive ion etching (RIE)
Chuanlong Guan et al.	2022	[114]	Ultraviolet nanoimprinting technology (UV-NIL)	Integrated polarization sensor	Sensor calibration method based on discrete Fourier transform and relative entropy
Kiyotaka Sasagawa et al.	2022	[115]	0.35 micron standard CMOS technology	Integrated polarization sensor	An optical system with a two-layer structure, including an external polarizer and a polarizer on a pixel array
Hongxia Zheng et al.	2022	[116]	65 nm 1P8MCMOS process	Integrated polarization sensor	Metal grid structure with dual operation modes

## Data Availability

Not applicable.

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
