# Peer review of "Biomimetic Polarized Light Navigation Sensor: A Review"

_sensors, 2023, doi:10.3390/s23135848_

Round 1

Reviewer 1 Report

Thanks for inviting me to review the paper entitled “Biomimetic Polarized Light Navigation Sensor: A Review”. In this paper, a systematic and in-depth review of polarized light navigation sensors is carried out. The paper can be accepted after the following issues are addressed.

1.The introduction to the significance and background of the study on bionic polarized light navigation sensors is not detailed enough, and relevant literature and application cases should be supplemented.

2. Most of the review comments are just digest of the research works done by the authors. The performance of different sensors should be supplemented when necessary.

3. The conclusion and outlook part is not attractive. Authors should supplement this part in more detail. Such as analyze per imaging sensors what the state of the art in knowledge is, what are the biggest uncertainties as mentioned in the literature, which imaging sensors are proven? How to proceed in this field?

4. Some pictures are not clear enough, and may need to provide a version with a higher resolution. Such as Fig(2), Fig(10), et al.

5. Please add spaces between numbers and units.

6. Please check the affiliation in page 1.

Reviewer 2 Report

The main question addressed in the reviewed manuscript is to collect, review and classify recent results in biomimetic polarized light sensors as well as to evaluate the progress made in this field.

The evaluation of the review paper presents a slightly different task than the assessment of an usual scholar contribution. The review paper should meet certain specific conditions, such as addressing a well-defined topic, obviously better if of interest and importance. Undoubtedly these conditions are fulfilled in the present case since the topic of biomimetic polarized light navigation sensors has attracted great interest in recent years.

At the beginning authors present a brief discussion of the parameters measurement of polarized light and the resulting possibility of determining the direction of movement based on the biomimetic approach. Next, they propose a classification of polarized light navigation sensors. According to the proposed scheme three main families of the sensors can be distinguished: point source polarized light sensors, imaging polarized light sensors and focal plane light splitting sensors. Every of these subsets is further divided into more particular cases. The presentation is comprehensive and provided with plenty of references that well reflect the collected results and present state-of-the-art. The whole paper seems to me to be well organized and correctly presenting the present status of biomimetic polarized light navigation sensors.

What makes me to feel a little uneasy is that authors pay surprisingly little attention to the presentation of solutions that can be found in biology and serve as an inspiration in sensor design. Exactly speaking only one sentence in the introduction and one reference cited in the 12th place in the manuscript (and here as Ref. [1]) can be taken as a direct quotation. By the way, this publication has its predecessors worth mentioning [2]. It seems to me that presence of the word biomimetic in the title of the manuscript would require more attention paid to this issue, especially given the wealth of examples, including polarized moonlight as an aid to orientation [3], also existing in the designed device [4].

The authors are right writing that the topic is hot and many papers have appeared recently. Among them also numerous review papers have been published. Lack of reference to the latter ones as well as lack of comparison of the present manuscript with them I would consider as another omission of the paper what is even more surprising taking into account that the number of references in the paper is quite long and reaches almost 100 positions. Let me add that some of these papers have appeared in the Sensors journal earlier, which requires even more reference and commentary [5-7].

Concluding, the manuscript seemed to me interesting and worth publishing, nevertheless, firstly it seems to me that authors could add a comparison of their work to earlier review articles showing novelty of their treatment of the topic, and secondly, mention in more detail the biological organs that are the source of inspiration for the solutions discussed in the paper.

References:

1.              M. Collett, T. Collett, S. Bisch, and R. Wehner, "Local and global vectors in desert ant navigation," Nature 394, 269–272 (1998).

2.              A.G. Dyer, A.D. Greentree, J.E. Garcia, E.L. Dyer, S.R. Howard, and F.G. Barth, "Einstein, von Frisch and the honeybee: a historical letter comes to light," J. Comp. Physiol. A 207, 449–456 (2021).

3.              J.J. Foster, J.D. Kirwan, B. el Jundi, J. Smolka, L. Khaldy, E. Baird, M.J. Byrne, D.-E. Nilsson, S. Johnsen, and M. Dacke, "Orienting to polarized light at night – matching lunar skylight to performance in a nocturnal beetle," J. Exp. Biol. 222, jeb188532 (2019).

4.              Y. Yang, Y. Wang, L. Guo, B. Tian, J. Yang, W. Li, and T. Chen, "Bioinspired polarized light compass in moonlit sky for heading determination based on probability density estimation," Chinese J. Aeronaut. 35, 1-9 (2022).

5.              S.B. Karman, S.Z.M.Diah, and I.C. Gebeshuber, "Bio-Inspired Polarized Skylight-Based Navigation Sensors: A Review," Sensors 2012, 14232-14261.

6.              G. Han, X. Hu, J. Lian, X. He, L. Zhang, Y. Wang, and F. Dong, "Design and Calibration of a Novel Bio-Inspired Pixelated Polarized Light Compass," Sensors 2017, 2623.

7.              J. Dupeyroux, S. Viollet and J.R. Serres, "Polarized skylight-based heading measurements: a bio-inspired approach," J. R. Soc. Interface 16, 20180878 (2019).

The article reads fluently, and since English is not my native language, in my opinion it means that English is good (at least for foreigners).

Round 2

Reviewer 1 Report

They responded well to my questions.